# Functional and structural reorganization in brain tumors: a machine learning approach using desynchronized functional oscillations
Joan Falcó-Roget [1] ✉, Alberto Cacciola [2], Fabio Sambataro [3] & Alessandro Crimi [1,4] ✉

Neuroimaging studies have allowed for non-invasive mapping of brain networks in brain tumors. Although tumor core and edema are easily identifiable using standard MRI acquisitions, imaging studies often neglect signals, structures, and functions within their presence. Therefore, both functional and diffusion signals, as well as their relationship with global patterns of connectivity reorganization, are poorly understood. Here, we explore the functional activity and the structure of white matter fibers considering the contribution of the whole tumor in a surgical context. First, we find intertwined alterations in the frequency domain of local and spatially distributed resting-state functional signals, potentially arising within the tumor. Second, we propose a fiber tracking pipeline capable of using anatomical information while still reconstructing bundles in tumoral and peritumoral tissue. Finally, using machine learning and healthy anatomical information, we predict structural rearrangement after surgery given the preoperative brain network. The generative model also disentangles complex patterns of connectivity reorganization for different types of tumors. Overall, we show the importance of carefully designing studies including MR signals within damaged brain tissues, as they exhibit and relate to non-trivial patterns of both structural and functional (dis-) connections or activity.

Functional networks of brain tumor patients, as obtained from functional magnetic resonance imaging (fMRI), show clear altered patterns of local and global disconnections[1] that follow functional rather than spatial distance to tumor location[2]. Consistent with these findings, a separate study also reported functional abnormalities that overlap with unaffected structural areas[3]. In addition, large-scale theoretical models of functional activity demonstrated differences in inhibitory connections between networks having similar structural features[4,5]. However, all these findings do not address the question of how the functional signal itself is modified by the presence of the tumor nor how it relates to potential desynchronizations in resting-state networks.

The oscillatory nature of functional time series has been widely exploited to entangle brain regions involved in a wide range of cognitive scenarios[6]. These functional co-activations, or connections[7], are computed by assessing temporal correlations which, by mathematical construction, rely on the oscillatory frequencies present in the signals. We hypothesized that the level of participation of each of those frequencies would be severely modified by the presence of a brain tumor. If this were true, we could quantify how these deviations propagate, or at least correlate, to alterations in the functional connections of resting-state networks similar to what was shown for a system of coupled oscillators[8,9]. Thereupon, we analyze local and brain-wide Fourier-transformed functional time series and study how their power spectrum singularities account for network anomalies in brain tumor patients.

A parallel line of research has recently focused on the use of diffusion magnetic resonance imaging (dMRI) in a wide variety of brain diseases[10,11]. Despite promising progress in the use of dMRI to resolve brain tumor microstructure[12], tumoral tissue is usually contaminated with cerebrospinal

[1]Brain and More Lab, Computer Vision, Sano Centre for Computational Medicine, Kraków, Poland. [2]Brain Mapping Lab, Department of Biomedical, Dental Sciences and Morphological and Functional Imaging, University of Messina, Messina, Italy. [3]Department of Neuroscience, University of Padova, Padua, Italy. [4]Faculty of Computer Science, AGH University of Krakow, Kraków, Poland. ✉e-mail: j.roget@sanoscience.org; a.crimi@sanoscience.org

fluid and gray matter abnormalities, therefore posing a challenge to existing fiber reconstruction methods. Although several intra-lesion fiber tracking methods have been developed, their usage is still scarce. Successfully removing the contribution of cerebrospinal fluid requires the use of low angular resolution tensor diffusion models unable to resolve complex white matter regions[13,14], or disregarding multi-shell acquisition schemes which can improve fiber orientation estimation when employing for instance constrained spherical deconvolution approaches[15]. Arguably, the lack of acceptance may be built upon the detrimental effects of disregarding the aforementioned MRI acquisition protocols and/or diffusion signal modeling approaches[16], due to a longer acquisition time that is often not suitable in the clinical setting. We propose and describe a hybrid pipeline using a single-shell-3-tissue algorithm[17] only in the tumoral area of the brain. Afterward, we merge the resulting connectivity matrix with the one obtained using a multi-shell algorithm in healthy tissue, exploiting the best features of each method and minimizing the impact of their downsides.

Unlike functional network studies, the knowledge of structural networks in brain tumors is, to some extent, unclear. Previous groups failed to find significant tumor-dependent differences between patients and healthy subjects[4,18], suggesting that network integration and segregation are mostly preserved. However, gliomas significantly altered the structural topology of the ipsilesional but not the contralesional hemisphere[19]. In addition, small structural differences vanished after surgery[5], implying that high-precision surgical interventions allow for spontaneous recovery of canonical organization.

A promising line of network neuroscience, commonly referred to as Spectral Graph Theory, builds on the idea that structural connections might be the source upon which functional activity and behavior rely on[20]. Likewise, structural constraints can be used to model hemodynamics while potentially revealing effective connections[21]. Yet, the bonds between structural and functional connections remain, not surprisingly, an open problem[22,23]. Unveiling mechanisms of structural plasticity evolution is, therefore, a key step toward understanding the impact of disruptions and recovery in both structural and functional connectomes. In this direction, and as a last contribution, we present a machine learning prediction study on how structural connections self-arrange after surgical resection of brain tumors.

Nonetheless, a certain number of challenges need to be addressed. Brain tumors, as well as their resections, critically affect elements of the network that may be far away from the damaged region itself[18]. Notwithstanding their success in representation learning and generative problems[24,25], graph neural networks struggle with complex network topologies[26] due to difficulties in aggregating information from long-range connections. Dehmamy and colleagues showed how this can be bypassed by designing modular networks. However, this path inevitably leads to complex data-hungry architectures[27] that cannot be trained in sensitive clinical scenarios where data is scarce. Interestingly, fully connected layers, besides being easier to train, naturally combine knowledge regardless of neighbor proximity[26,28]. Thus, we build on the idea that healthy structural connections could be used to inform and guide predictions. We propose to use anatomical constraints in a Bayesian framework combined with fully connected layers to produce detailed graphs that share both visual similarities and network topological characteristics with the ground truth.

In summary, we present several contributions in both functional and diffusion MRI domains in the presence of a brain tumor. Initially, we study BOLD signals within the tumor and how they relate to abnormal patterns of brain-wide resting-state connectivity and complexity. Secondly, we present our attempt to consider fiber tracking within the lesion combined with whole-brain tractography. Lastly, we study how the fibers in the lesion, as well as surgery, impact the structural connectivity after tumor resection.

## Results
### Tumor and default mode network functional signals
We first addressed the question of whether functional signals were present inside the lesion. The segmented masks included all parts of the tumor, from

the necrotic issue to the edema (see Methods). For this study, resting-state fMRIs were available (21 controls and 25 patients). We extracted functional signals from inside the lesioned parts of the brain and compared them with functional activity from the same region in control subjects (Fig. 1a, see also Methods). There were no straightforward differences concerning Blood-Oxygen-Level-Dependent (BOLD) signals from regions belonging to the Default Mode Network (DMN) in the control group (Fig. 1b, see also Methods).

We were interested in characterizing the relationship between BOLD oscillations present inside the masked lesion and brain areas known to be functionally active and coherent (i.e., belonging to DMN regions) in resting conditions[29]. Thus, to test whether and how functional activity inside tumors was related to global signals, we first studied how brain tumors themselves shaped those global signals. Direct comparison of averaged time series was not possible since arbitrary dephasings introduced artifacts (see Fig. S1a). Therefore, we analyzed functional series from regions belonging to the DMN in the frequency domain (see Methods). Alterations in the total power as well as the distribution of such power across frequencies were present in some subjects but not in others (Fig. 1c; see also Figs. S2–5a). The autocorrelation functions between time signals exhibited similar shapes for short time lags (Fig. 1b inset). We observed that left-skewed power distributions tended to have higher autocorrelations than right-skewed power distributions (Fig. S4c inset) suggesting slower time series maintained temporal coherence for longer times. For longer lags, however, all signals lost this coherence. This inspired the definition of a score capable of distinguishing patients displaying faster oscillations from patients who showed the opposite. We will expand on this idea in the next section.

A similar phenomenon was observed when pair-wise correlations between DMN regions were computed to reconstruct the network (Fig. 1d). Extensive research in network measures allowed us to estimate the complexity of networks based on the distribution of the correlations (see Methods). Briefly, we devised the Θ Richness score as the difference, in the module, between the distribution of correlations building the network and a uniform distribution[30]. Differences in the Θ Richness between patients and healthy networks were also found to be inconsistent across subjects (Fig. 1e; see also Figs. S2–5c). As a final step, we inspected signals from the same regions belonging to the DMN of both patients and control subjects, again finding no clear traces of tumoral damage in DMN functional signals (Fig. 1f).

In summary, functional signals from tumors and DMNs did not show significant changes in terms of complexity. However, they displayed alterations both in the power domain and in the distribution of functional connections.

### Temporal dynamics and default mode network reorganization
To further characterize DMN signals in brain tumors bypassing phase and noise artifacts (perhaps unsuccessfully removed by the preprocessing pipeline), we designed a scalar score based on the cumulative power distribution of the time series. This Dynamics Alteration Score (DAS), inspired by previous work on periodic modeling of fMRI time series[31], was able to differentiate between slower or faster oscillations of BOLD signals (see Methods; see also Fig. S1). Overall, the dynamics of DMN in patients were found to be positive, negative, or zero. We measured the similarity of the patients and control networks with a node similarity score (see Methods) and observed that it is significantly anti-correlated with the magnitude of the DAS ($r = -0.506$, df = 23, $p = 0.01$, two-tailed exact test; Fig. 2a). We studied the similarity as a function of the direction in the alteration expecting it would decrease regardless of the change in dynamics for large scores. Indeed, we found a clear inverted "U-shaped" pattern that agreed with the interpretation that altered dynamics is a contributing factor in network reorganization (Fig. 2a Inset). Negative DAS was positively correlated with similarity ($r > 0$, $p = 0.006$, df = 11, one-tailed permutation test) while positive DAS exhibited the opposite trend ($r < 0$, $p = 0.083$, df = 14, one-tailed permutation test).

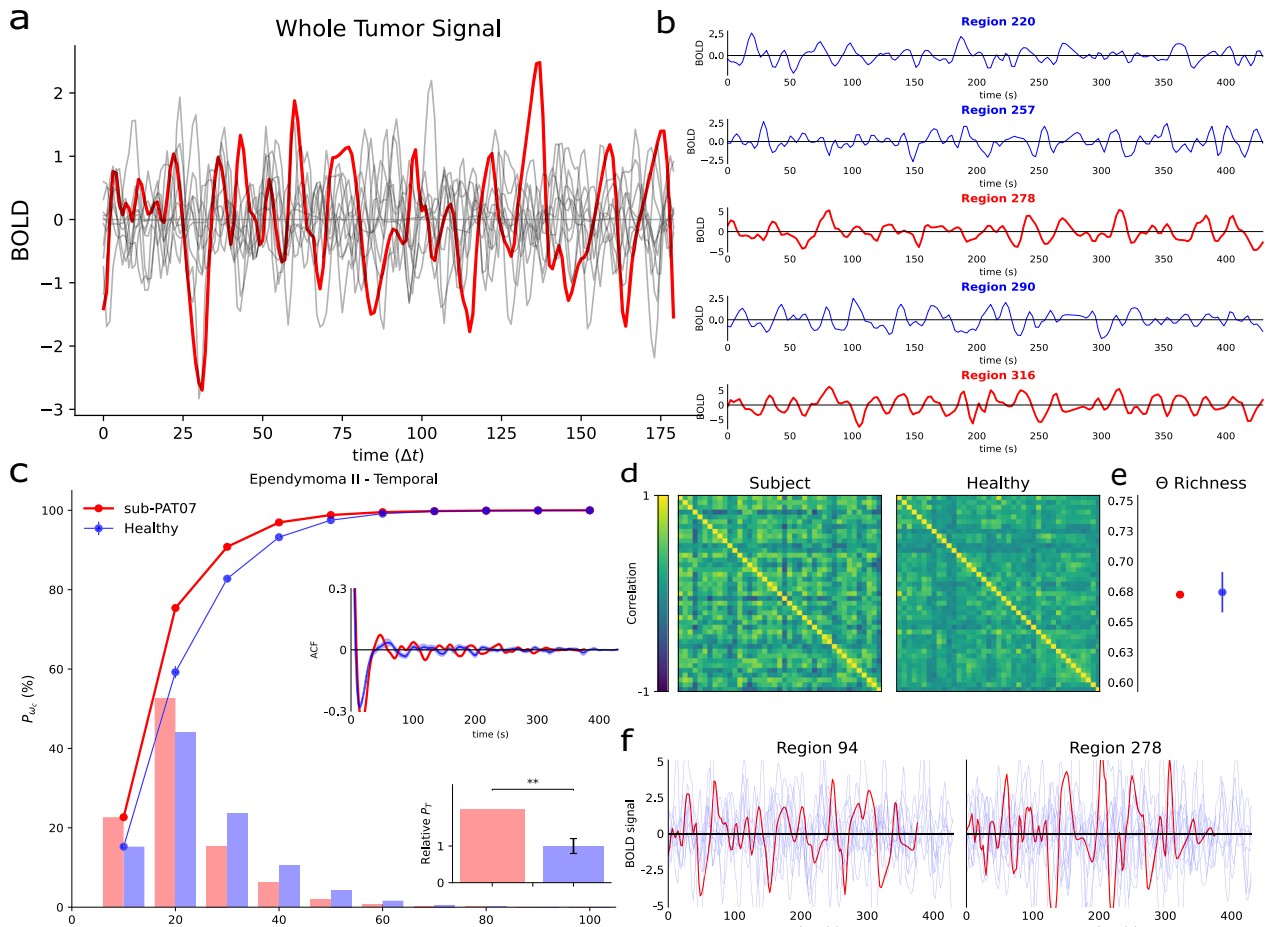

**Fig. 1 | sub-PAT07 Edema and DMN functional signals. a** Mean functional signal (BOLD) measured inside the edema of a patient (red) as compared to signals from the same region in healthy subjects (gray). Similar signals were also found in all subjects inspected. The time axis is shown in time steps rather than seconds. **b** Functional signals from 5 example regions belonging to the DMN of control subject 10 (randomly selected). Colors code for regions assigned to the same community through the Louvain Community detection algorithm. No qualitative difference is found between the raw time series inside the edema of a patient and regions of the DMN of a healthy subject. **c** Cumulative power (lines), power distribution (bars), autocorrelation function (ACF), and total power relative to healthy subjects (histogram) for the patient (red) and mean of controls (blue) functional signals of the regions in the DMN. Error bars [mean ± SEM] indicate the results for the control population, while "*" codes for statistical significance. The relative power was obtained by averaging over subjects and DMN regions and taking healthy subjects as the baseline (i.e., equal to 1). **d** Functional DMN from the same patient and mean of control subjects (see Methods). **e** Functional complexity as measured by the distribution of correlations for the patient's network (red) and the mean of control (blue) [mean ± SEM] (see Methods). **f** Functional signal of two randomly selected regions from the same patient (red) as opposed to all the control subjects (light blue). No apparent difference is found between raw time series across regions, subjects, and patients.

Additionally, we found a small but strong difference in the magnitude of the change in Θ Richness between healthy and brain tumor networks ($|\Delta\Theta| = 0.086$, $p < 0.001$, df = 23, one-tailed $t$-test; $p = 0.054$, one-tailed U-test). Opposite to this, the direction of this change (i.e., considering increases or decreases) was non-significantly different than zero ($p = 0.067$, df = 23, two-tailed $t$-test; $p = 0.286$ two-tailed U-test). Negative DAS implied the presence of higher frequencies of oscillations challenging the existence of coordinated oscillations. Consistent with this, alterations in the BOLD dynamics significantly correlated with changes in functional complexity ($r = 0.514$, df = 23, $p = 0.009$, two-tailed exact test; Fig. 2b). An equivalent trend appeared when considering absolute alterations ($r = 0.413$, df = 23, $p = 0.04$, two-tailed exact test).

In conclusion, we found that changes in the signal oscillations of the DMN regions translated into significantly different patterns of functional connections. Slower signals increased the complexity of the patients' network while faster oscillations decreased it. Crucially, the appearance of any sort of disturbance in the BOLD dynamics was associated with a poorer similarity with the DMN from healthy subjects. In the following section, we explore how and where these alterations may arise in brain tumor patients.

## Local and distributed functional signals arising from brain tumors

We asked whether spatial proximity between the tumor and the DMN could explain desynchronization. The magnitude of alteration in the dynamics of DMN was not correlated with the mean Euclidean distance ($p = 0.583$, df = 23, two-tailed exact test) nor with the total overlap ($p = 0.29$, df = 23, two-tailed exact test) between tumor and DMN centroids (Fig. 2c).

Next, to assess whether intra-tumor functional activity was both existent and relevant, we compared the DAS computed from the DMN with that obtained from intra-lesion signals (see Methods). The alterations in the signals from the tumor and the DMN of the same patient were highly and significantly correlated ($r = 0.696$, df = 23, $p < 0.001$, two-tailed exact test; Fig. 2d). This result ultimately linked what happened inside the tumoral tissue with what was observed across spatially distributed brain regions. That is, abnormalities in the signals were shared between lesioned and unaffected regions. We also compared the properties of the BOLD signal inside the tumor with the same regions in healthy subjects. Alterations in lesioned areas were observable in the power of the signal ($|P| > 0$, $p = 0.031$, df = 23, one-tailed $t$-test) but they were significantly more pronounced in the dynamics (Fig. 2e; $|DAS| > 0$, $p < 0.001$, df = 23 one-tailed $t$-test).

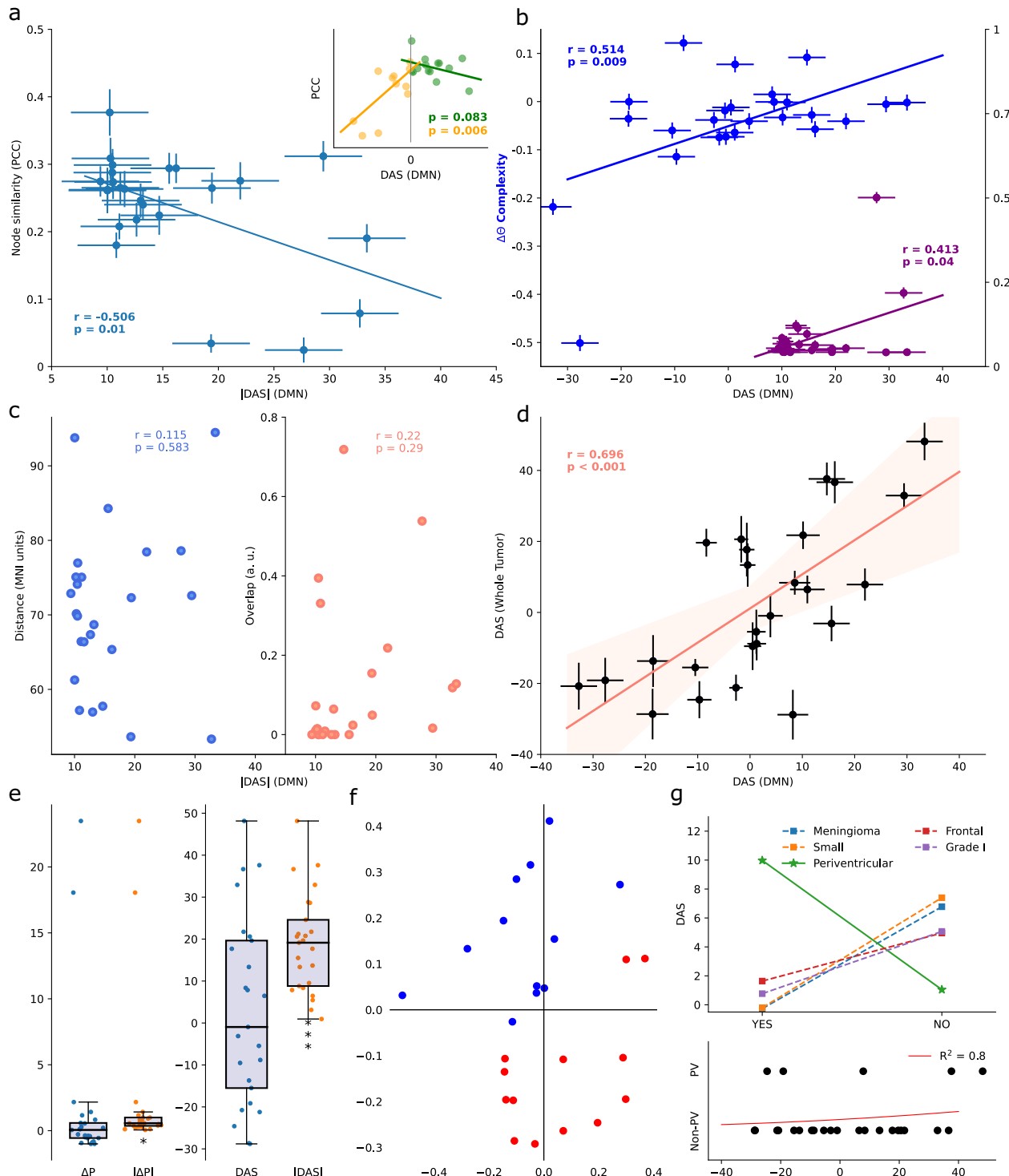

**Fig. 2 | Linking DMN and tumor BOLD signals.** Pearson Correlations and the corresponding *p* values are shown. Error bars correspond to the mean ± SEM. **a** Correlations between node similarity as measured by the Pearson correlation coefficient and the Dynamics Alteration Score (DAS) from the Default Mode Network (DMN). The inset shows how alterations in the dynamics shape the organization of the network regardless of the direction of the displacement of the frequency spectrum. **b** Alterations in dynamics correlate with alterations in network complexity. Change in the complexity of the patients' DMN with respect to the healthy pool (Directionality ΔΘ and Magnitude |ΔΘ|) as a function of the DAS. **c** Scatter plot showing the null correlation of DAS with distance and overlap between lesion and the DMN centroids. **d** A strong linear trend was found between alterations inside the patients' tumor and alterations in the DMN as measured by the DAS. The orange line

corresponds to the significant linear fit (two-tailed exact test), and the shaded areas mark the 95% confidence interval. **e** Differences in total power relative to healthy subjects (LEFT) and dynamics (RIGHT). Only the absolute values were significantly different from zero. **f** Scatter plot of the two components found via Fast Independent Component Analysis applied to the DAS differences between patients and control subjects. The same two clusters (red, and blue) were consistently found with a K-means score of -1.2. **g** TOP, Alteration in dynamics between different groups of tumors. BOTTOM, Logistic regression between DAS and periventricular (PV) tumor patients. Qualitative inspection reveals a tendency for periventricular tumors to display slower BOLD dynamics, although they were found to be non-significant presumably due to the small number of samples ($N = 5$, $p = 0.31$, one-tailed U-test).

As opposed to changes in network organization and complexity (Fig. 2a), there was no statistical relationship between alterations in the intensity and the oscillations ($r = -0.113$, df = 23, $p = 0.588$, two-tailed exact test).

Dimensionality reduction via Fast Independent Component Analysis was applied to cluster differences between patients and control subjects. The same two clusters were consistently found with a K-means score of ~ −1.2, although no apparent groups were observable (Fig. 2f). As such, we hypothesized that the alteration in power distributions might be explained by tumor features. We divided subjects into two groups according to 5 tumor features: location (brain lobe), size (with respect to the sample median), histology (based on histological features of the resected tissue), grade (based on the cell appearance and differentiation in the sampled tumor, which reflect tumor aggressiveness, i.e., the ability to grow, infiltrate and recur) and, periventricular (when located at the borders of the ventricles). If the tumor intersected the ventricles, we expected the cerebrospinal fluid to penetrate the lesion thus severely disturbing and slowing the signal recovered. Tumors intersecting the ventricles showed a large positive DAS, although non-significant (Fig. 2g TOP; $p = 0.31$, one-tailed U-test). However, given its larger magnitude and greater significance with respect to all other groups, we provide a possible explanation of its meaning in the discussion. A logistic regression also displayed a reasonable determination coefficient and pattern ($R^2 = 0.8$), but the small number of periventricular tumors did not allow for further quantitative assessment (Fig. 2g BOTTOM).

As a final note, we report that the tumor with the highest DAS was a grade II ependymoma located in the temporal lobe and intersecting with the left lateral ventricle. The tumor with the lowest (and negative) DAS corresponded with a grade I meningioma located in the frontal-skull base not intersecting the ventricles.

## Structural brain networks containing intra-tumoral and peritumoral fibers

As a first step towards reliable fiber tracking inside brain tumors, we designed a combined approach that used two different (previously validated) constrained spherical deconvolution fiber orientation function (FOD) reconstruction algorithms (Fig. 3a). As suspected[15], the multi-shell approach caused overdamping of the diffusion signal within the lesion mask (Fig. S6). In combination with this, the altered anatomical properties surrounding the lesion imposed an artificial early stop to streamlines that could, if diffusion signal was available, penetrate or circumvallate the tumor (Fig. 3b; see also Supplementary Material). Instead, the careful combination of FODs and the relaxation of anatomical constraints in the peritumoral tissue allowed our pipeline to reconstruct well-known white-matter pathways that were otherwise truncated (Fig. 3c). Importantly, these emerging tracts appeared as natural extensions of those stopped by the lesion.

However, validation of any fiber tracking pipeline is difficult due to the lack of a unique ground truth. Nonetheless, we inspected the scale-free properties of the weighted degree distributions of the reconstructed networks. For every healthy subject and patient, we fit the tail of these distributions to a power law, characterized by a single exponent $\alpha$ (see the Supplementary Material). Distributions showed scale-free properties with acceptable Kolmogorov-Smirnov distances for different scaling ranges confirming a power law organization in the asymptotic limit ($\alpha \in [2, 3]$, $D \in [0.15, 0.22]$, $\sigma \in [0.09, 0.62]$). Crucially, differences in the power law distribution of the healthy networks compared to the lesioned networks obtained with our pipeline were comparable to the differences of the networks obtained with a canonical multi-shell multi-tissue pipeline (Fig. S6d; $p > 0.05$, two-tailed Wilcoxon test, two-tailed U-test). Yet, the differences in the distributions between our hybrid approach and the multi-shell multi-tissue one increased with tumor size. An ordinary linear squared model showed a significant correlation ($R^2 = 0.439$, $F = 5.484$, $p = 0.006$) between alterations in the power law behavior and several tumor features. Tumor size was the most significant predictor ($p = 0.005$, df = 21, two-tailed $t$-test). Thus, differences between our pipeline and the standard approach disappeared with small tumors (Fig. S6).

## Structural predictions after tumor resection

One of our main goals was to design a method capable of predicting how structural graphs will look after major surgical procedures while still elucidating the confounding effects of connectivity reorganization. Previous work showed that linear models could surprisingly capture the fundamental properties of structural graphs[32]. We evaluated a Fully Connected NETwork (FCNET) against a Huber Regressor and a null model. The choice of a Huber Regressor was motivated by its robustness to outliers and the presence of heterogeneous data points. Testing against null models helps in avoiding circular analysis[33], therefore we also benchmarked FCNET against an untrained linear generator. Both the outputs of the Huber and null generators were weighted by the same anatomical prior as the FCNET. For each model and fold, we tested the left-out network with 6 different metrics (see Methods). The results for each score are shown in Table 1 for the mean and in Table S1 for each subject in the dataset.

FCNET significantly outperforms the null model in all evaluation metrics ($p < 0.001$ two-tailed U-test). When tested against the Huber regressor, FCNET outperformed it in the metrics assessing numerical reconstructions ($p < 0.05$ two-tailed U-test; for MSE and MAE) as well as metrics assessing similarity ($p \sim 0.05$ two-tailed U-test; for PCC and CS). However, when tested for topological accuracy, FCNET did not improve with respect to the Huber regressor measured by the Kullback-Leiber (KL) and Jensen-Shanon (JS) divergences ($p > 0.4$, two-tailed U-test). Despite not being trained on preserving topological features, both FCNET and Huber captured structural properties since both models significantly decreased the KL ($p < 0.001$, $F > 300$, one-way ANOVA; $p < 0.001$, $F > 30$, Kruskal-Wallis test) and JS ($p < 0.001$, $F > 300$, $df_{between} = 2$, $df_{within} = 54$, one-way ANOVA; $p < 0.001$, $F > 30$, Kruskal-Wallis test) divergences of the weight probability distributions between predicted and ground truth networks (see Methods).

None of the models was trained using a regularization method to prevent negative connections. Surprisingly, however, FCNET generated negative connections that accounted for less than 25% and they were all between 0 and −0.5. Since these values are in log scale, they would account for a connection of less than 1 and get filtered by the anatomical threshold. We show the generated post-surgery networks and residuals of three randomly selected subjects in Fig. 4.

## Subject-specific predictions

The connections within brain networks are highly heterogeneous in the context of a brain tumor. Herein, we imposed anatomical prior to act as a regularization method. However, a highly restrictive prior resulted in a complete loss of subject specificity despite FCNET achieving lower reconstruction errors. After some trials, we found that an optimal (or nearly optimal) prior was able to discard enough connections while still capturing some inter-subject variability of the networks (Fig. 4 red squares; see also Fig. S7). However, model generalization does not allow for a perfect fit to all data points (Fig. 4 right column).

Next, we asked whether FCNET was simply overfitting a small subset of similar subjects. We calculated the z-score of each metric with respect to the 19 folds cross-validated subjects (see Methods). For all metrics, we found that ~65% of all z-scores lied in the $\pm\sigma$ range and ~95% fell in the $\pm2\sigma$. Even more, when repeating the training with different starting weights, all subjects but 2 showed different scores (Fig. 5a); they were not uniquely classified as outliers and therefore were considered in all the subsequent analysis ($p > 0.05$, df = 17, two-tailed Grubbs test). As a further checkpoint to ensure that the model was not overfitting to a specific subset of similar subjects (i.e., patients with similar tumors), we tested for the normality of the z-scores, thus finding that all of them had a significant linear correlation ($r > 0.9$, $p < 0.01$ two-tailed $t$-test) between the theoretical and observed quantiles (Fig. 5b).

We used four metrics to evaluate the model's output measured the numerical similarity between the ground truth and predictions. Not surprisingly, they exhibited high pair-wise correlations ($|r| > 0.8$ and $p < 0.01$, two-tailed permutation test), therefore post hoc analyses on one score were generalizable to all. Unlike MSE and MAE, both the PCC and

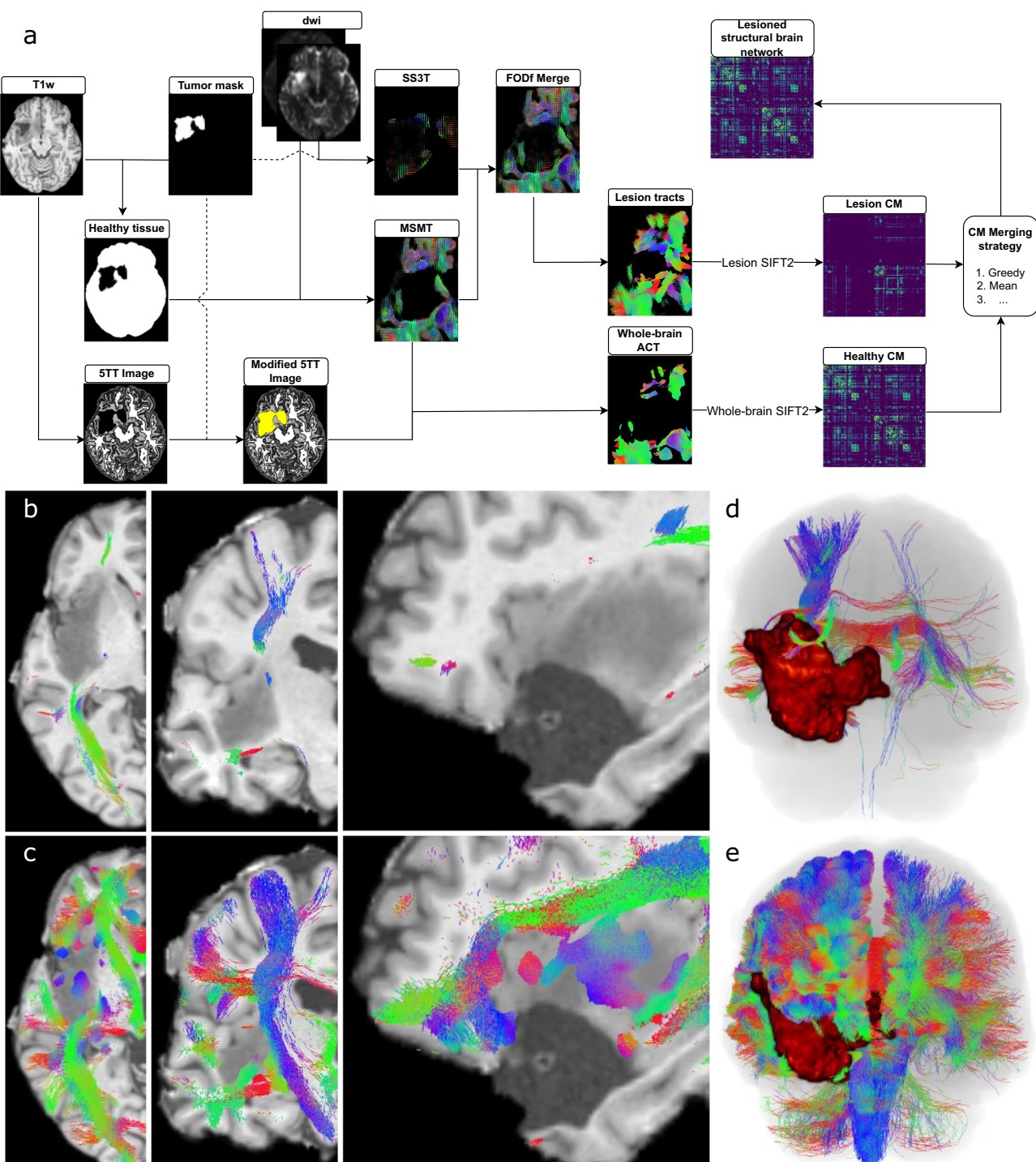

**Fig. 3 | Reconstruction of brain structural networks. a** Summary of the workflow designed to generate the tractograms in the presence of a brain tumor. Multi-shell multi-tissue (MSMT) reconstruction methods are run together with anatomically constrained tractography (ACT) to obtain the "healthy" fibers without including lesioned tissue. Fiber orientation functions inside the lesions are obtained by running the single-shell 3-tissue (SS3T) method only inside the lesioned regions (Tumor mask). Fiber orientation functions (FODs) from both methods are then merged. Connections originating, terminating, or traversing oedemic tissue are tracked with the iFOD2 probabilistic algorithm by only seeding inside the lesion. A maximum angle and FOD amplitude cutoff stopping criteria are used. Both tractograms are used to obtain a weighted connectivity matrix with the outputs of streamline filtering

(i.e., SIFT2). As a final step, the *lesion* and *healthy* structural matrices are merged using a customized formula (e.g., greedy). **b** Axial, coronal, and sagittal planes (thickness of 1 mm) of the tractogram from sub-PAT16 obtained with a simple multi-shell multi-tissue without masking the tumor (see Methods). 1 mm thick cropping point is (x,y,z) = (35,−17,−3) mm in MNI space. **c** Tractogram from sub-PAT16 identical coordinates and thickness as in (**b**) but having used the hybrid method outlined in (**a**). Importantly, large cortico-spinal and superior longitudinal fasciculus peritumoral fiber bundles are now visible. Overview of the whole brain tractogram of the same subject with a simple MSMT approach (**d**) and our hybrid pipeline (**e**).

**Table 1 | Model results (mean ± SEM)**

| Model | MSE | MAE | PCC | CS | KL | JS |
|---|---|---|---|---|---|---|
| FCNET | **1.91 ± 0.07** | **0.83 ± 0.01** | **0.807 ± 0.008** | **0.859 ± 0.006** | **9.09 ± 0.12** | **0.68 ± 0.01** |
| Huber | 2.14 ± 0.07 | 0.87 ± 0.02 | 0.785 ± 0.007 | 0.841 ± 0.006 | 9.18 ± 0.13 | 0.69 ± 0.01 |
| Null | 8.09 ± 0.17 | 1.61 ± 0.03 | 0.004 ± 0.002 | 0.005 ± 0.003 | 13.13 ± 0.07 | 0.814 ± 0.003 |

The Fully Connected NETwork (FCNET) was tested using a Leave One Out cross-validation scheme in 6 metrics (MSE: Mean Squared Error; MAE: Mean Absolute Error; PCC: Pearson Correlation Coefficient; CS: Cosine Similarity; KL Kullback-Leiber and JS: Jensen-Shannon Divergences). In addition, we tested against two benchmark models: Huber and Null regressors. FCNET showed a significant improvement in all metrics evaluated. Both Huber and Null models were also tested in the same framework, that is, the predicted likelihoods weighted by the same anatomical prior (see Methods). Bold numbers indicate better performance; lower is better for MSE, MAE, KL, and JS while higher is better for CS and PCC.

**Fig. 4 | FCNET's network generation.** Three subjects were randomly selected to be displayed as visual proof that FCNET captures the essential properties of the post-surgery graphs. The residuals show the absolute difference between the predicted and ground truth networks. Negative connections were dropped for visualization purposes since they crucially affected the color scale but not the structure. FCNET can capture some specific inter-subject variabilities (augmented red squares) despite being trained on highly heterogeneous data. Connection strength is measured as $\log(1 + w)$ where $w$ is the native connectivity derived from the tractograms. PAT03: meningioma grade I, location parietal right, 78.44 cm³. PAT15: meningioma grade I, location frontal right, 2.13 cm³ PAT28: oligodendroglioma grade II, location frontal left, 11.49 cm³.

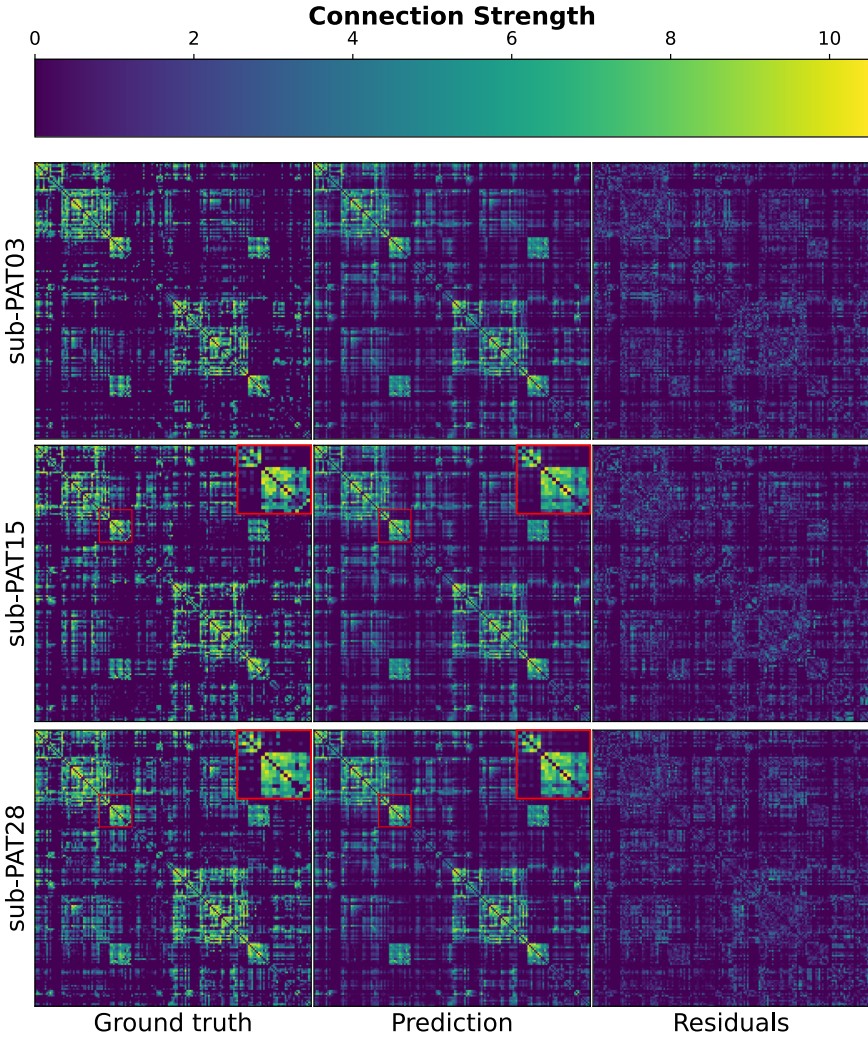

CS are normalized between -1 and 1 making it desirable regarding reproducibility. Without loss of generality, we based the following analysis on the PCC.

## FCNET's sensitivity to tumor size

Structural disconnections are typically correlated with the size of the lesion and cognitive deficits[34]; thus, we hypothesized that large tumors would decrease the performance of the network. Considering all subjects, correlations between accuracy and size were not found to be statistically significant (Fig. 5c; $r = 0.0183$, $p > 0.5$, df = 17, two-tailed permutation test). However, we redid the same analysis without the three largest tumors as they appeared to be abnormally large. Those tumors had volumes greater than 60 cm³ and their mean value was considered an outlier ($p < 0.01$, two-tailed Grubbs Test). In this scenario, correlations drastically increased both

in value and significance ($r = -0.336$, $p = 0.04$, df = 14, two-tailed permutation test); yet, when disregarding the fourth largest tumor, relative changes in correlations were less pronounced ($r = -0.451$, $p < 0.01$, df = 13, two-tailed permutation test).

To settle whether the effect of size was indeed present, we divided all patients, including the three largest ones, into two groups using the median of the whole dataset (Fig. 5d; P50 = 12.95 cm³). Subjects with small tumors (size < median) showed higher PCCs (0.825 ± 0.007 [mean ± SEM]) than subjects with large tumors (0.792 ± 0.013 [mean ± SEM]) ($p = 0.03$, df = 17, one-tailed $t$-test; $p = 0.03$, one-tailed U-test).

In conclusion, the highly non-linear generative model was sensitively worse when considering large tumors. However, there seemed to exist confounding effects distorting the relationship. In the next section, we explored those in more detail.

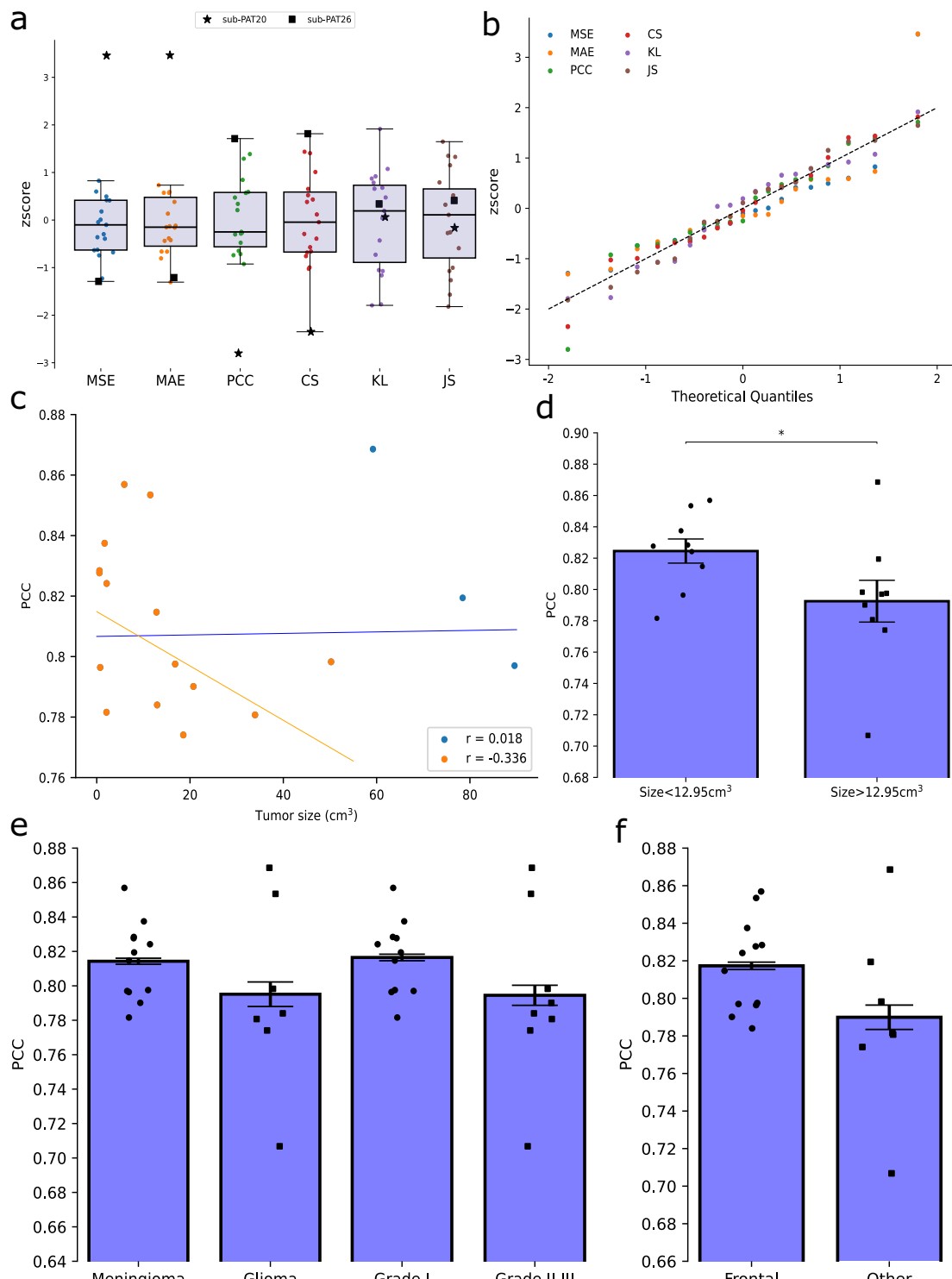

**Fig. 5 | FCNET's selective predictions. a**, **b** Normality assumptions on the whole pool of metrics used to test the model in each 1-fold (MSE: Mean Squared Error; MAE: Mean Absolute Error; PCC: Pearson Correlation Coefficient; CS: Cosine Similarity; KL Kullback-Leiber and JS: Jensen-Shannon Divergences). Boxplot of zscores in (**a**). Shaded areas represent the Inter Quartile Range (Q3–Q1) and the solid black line depicts the median. Marked subjects (PAT20 and PAT26) were not classified as outliers for all the metrics used ($p \sim 0.15$, Grubbs one-tailed test). Q–Q plot for each metric in (**b**). The dashed black line shows the expected $y = x$ relation. The significance for each regression was assessed ($r > 0.95$ and $p < 0.01$). **c**, **d** Impact of tumor size in the predicted graphs. Correlations between PCC and tumor size in (**c**) considering all but the three largest patients (orange) and considering all subjects (orange+blue). Differences in the PCC [mean ± SEM] between small and large tumors in (**d**). The subdivision was made based on the median of the set (P50 = 12.95 cm³). **e** Differences in the PCC [mean ± SEM] between tumor type and tumor grade ($p = 0.037$, one-tailed U-test). **f** Differences in the PCC [mean ± SEM] between tumor locations.

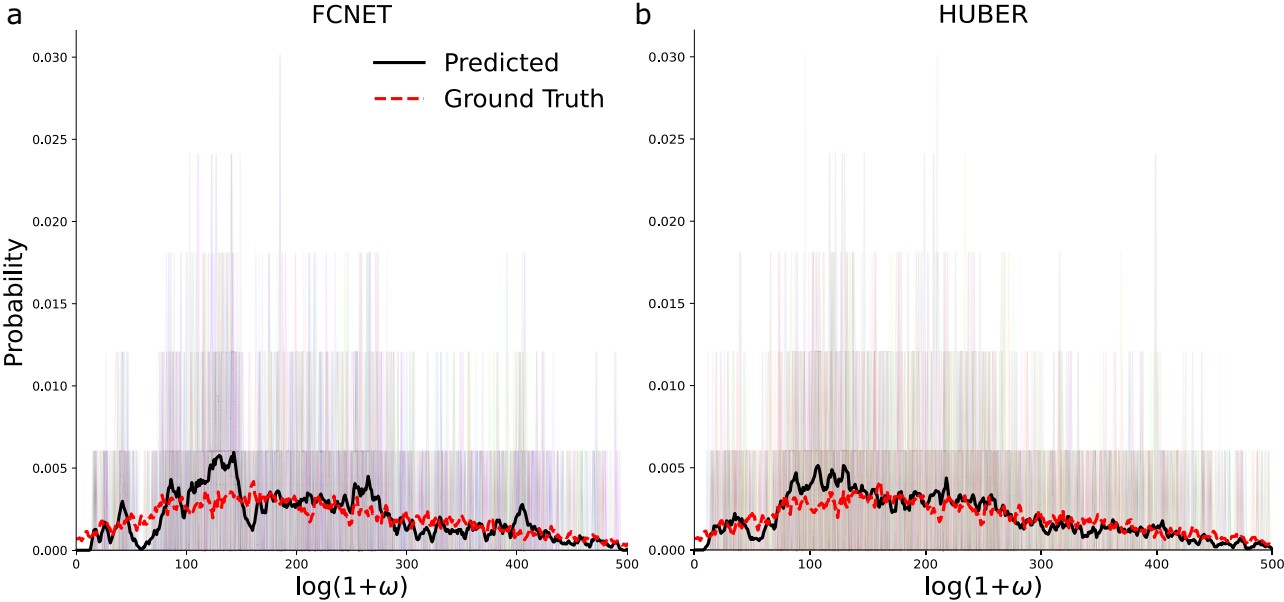

**Fig. 6 | FCNET's topological accuracy.** Black thick lines show the mean weight probability distribution predicted by FCNET (**a**) and the HUBER regressor (**b**). The dashed red line shows the mean weight probability distribution of the real post-surgery graphs. Shaded background bars show the predicted distributions for each subject in the dataset.

## Potential confounding effects on structural connectivity reorganization

The histology of the tumor greatly influences clinical considerations such as survival rates or possible treatment strategies. Meningiomas tend to exert pressure on the healthy brain tissues in opposite to gliomas which show more infiltrative behaviors. However, we did not find significant differences (Fig. 5e; $p = 0.29$, df = 17, two-tailed $t$-test; $p = 0.3$, two-tailed U-test) in the PCC between meningiomas ($0.814 \pm 0.002$ [mean ± SEM]) and gliomas ($0.795 \pm 0.007$ [mean ± SEM]). Predictions were slightly better for low-grade tumors ($0.816 \pm 0.001$ [mean ± SEM]) than high-grade ($0.794 \pm 0.003$ [mean ± SEM]) but without reaching statistical significance ($p = 0.15$, two-tailed U-test).

Since not all regions of the brain are equally important in terms of structural connectivity, we also tested if the location of the tumor had an impact on the predictions. The dataset included 12 patients with frontal and 7 patients with temporal or parietal tumors. We found differences in performance between frontal and non-frontal tumors (Fig. 5f). Despite not being significant ($p = 0.12$, two-tailed U-test), frontal tumors showed higher PCC ($0.817 \pm 0.002$ [mean ± SEM]) than temporal and parietal together ($0.789 \pm 0.003$ [mean ± SEM]).

Lastly, and contrary to what we observed in functional dynamics, the results of FCNET were not sensitive to whether the tumor intersected the ventricles ($p = 0.82$, two-tailed U-test).

## Topological predictions

We also tested the topology of the generated networks by computing the weight probability distribution. The loss function used to train all models did not have any topological term, but the generated networks shared global properties with the ground truth as measured by the KL and JS divergences in Table 1. The generated graphs showed a biological lognormal weight distribution with a small number of highly connected nodes (Fig. 6; see also Supplementary Materials).

## Discussion

The focus of the present work is to quantitatively study functional and diffusion MRI signals in the presence of brain tumors. Importantly, unlike previous studies, we focus on the signal present within the lesion and how it explains connectivity reorganization in both imaging modalities. We show that when functional time series are compared in the frequency spectrum, the distribution of oscillatory frequencies is strongly related to how resting-state connections are re-wired within the DMN. We also observe that the power spectrum within the brain tumor regions (including edema) is strongly related to an oscillatory decoupling between the distributed and active brain regions. Moreover, we designed a hybrid tractography pipeline capable of combining anatomical constraints and diffusion signal from the lesioned tissue. Finally, we use the reconstructed networks before and after brain surgery to train a machine learning model able to extract information about non-normative patterns of structural reorganization.

We also set out to determine whether functional activity was present within the brain tumor. We find qualitatively similar signals between healthy and tumor tissues. However, the fact that there is a non-zero tumor functional signal is not indicative of its relevance. Therefore, to explore the functional sanity of the signals found, we first look at the existing differences in DMN between patients and healthy controls.

Functional connections rely on temporal correlations between time signals which are entirely dependent on the phase as well as the active frequencies in the temporal signals[7]. Because of this mathematical construction, we hypothesized that the most descriptive feature of Blood-Oxygen-Level-Dependent (BOLD) time series would be the power distributions across frequencies. For this purpose, we designed a DAS that can capture not only how different two power distributions are, but also how these differences are related with the relative dynamics of those signals (see Methods; see also Figs. 1c and S1–S5). Intriguingly, the alteration in the signal dynamics (measured by the DAS), strongly correlated with network closeness (measured by node similarity; Fig. 2a).

The integrity of resting-state activity of the brain has been associated with cognitive function[6]. Based on this paradigm, numerous studies have explored how functional damage translates into cognitive impairment[1,3–5,35]. As a clear demonstration of the resilience of the brain to damage, all of them reported a continuous rather than binary decrease in cognitive performance. Brain networks undergo rearrangement to minimize the impact of lesions, and this rewiring follows specific constraints[36]. From electroencephalography recordings, Wolthuis and colleagues found the level of topological similarity to be predictive of post-surgery cognitive skills[35], suggesting once again that understanding how the brain responds to lesions is crucial to predict the impact of surgical procedures on cognitive function.

To this purpose, we computed the complexity of the DMN and found significant alterations in magnitude with variable directionality across patients. Approximately half of the patients display increased network complexity while the rest exhibited the opposite pattern (Fig. 2b). What these two non-contradictory findings suggest is that different tumors and/or patients react differently when considering DMN reorganization which inevitably leads to different network complexities. A strong positive trend exists between the presence of slower oscillations (DAS > 0) and higher complexities (Fig. 2b). We may speculate that faster time series resemble noisy temporal structures lacking temporal coherence, therefore, resulting in a more uniform distribution of functional connections. In contrast, for slower functional dynamics, more complex networks would naturally arise. However, as stated by the proponents of the measure used here[30], high complexity does not imply an optimal segregation-integration balance promoting small-world topologies and their subsequent advantages in information propagation[37].

We also try to answer the origin of these desynchronizations. Current knowledge states that structural closeness is not sufficient to understand disruptions in patterns of functional disconnections[1-3]. In accordance with this, changes in BOLD dynamics were not explainable by the spatial overlap nor distance between lesions and DMN (Fig. 2c). Interestingly, dynamic alterations in DMN regions increased rather linearly with the distance to the tumor, from faster oscillations in nearby tumors to slower ones in spatially separated lesions ($r = 0.442$, $p = 0.027$, two-tailed exact test). However, distance does not account for the difference in network rearrangements given that dissimilarity increases for both slower and faster dynamics.

Instead, DMN desynchronization is better explained by frequency shifts inside brain tumors (Fig. 2d). Current neuroscience methods struggle to address questions of causality, but for the sake of completeness and transparency of the results shown, we provide a speculative causal line to the findings reported. A plausible explanation would be to consider that the tumors themselves are altering the neural dynamics locally. Then, such alterations would propagate through spatially distributed brain regions, perhaps supported by structural connections[20,21,23,38], causing a global DMN desynchronization. To try and compensate, BOLD correlations between regions would rearrange themselves to spare the functionality. Network similarities and topologies across subjects would change, as found by previous studies[1-3,35], without greatly altering their complexity. However, DASs could not disentangle whether the tumor altered the signal and that alteration spread to the DMN or the opposite. The relationship between the alterations in the DMN and the tumor remained unchanged when considering only the tumor core or the non-necrotic tissue of the lesion for patients with gliomas (Figs. S8–S9).

Finally, periventricular tumors display a higher (positive) DAS than non-periventricular tumors. These differences do not reach statistical significance but are approximately five times larger in magnitude with respect to other groups. An increase in the volume of cerebral and lymphatic fluids inside the tumor may be the source of a larger desynchronization in the BOLD signal. Further research is required to determine whether this is associated with worse survival rates in higher-grade gliomas, such as glioblastomas[39,40]. Rather, tumors growing within the ventricles may result in obstruction of CSF flow leading to ventricular dilation and affecting CSF dynamics in ways that need not resemble the ones we reported.

Structural brain networks, commonly referred to as structural connectomes, should be reconstructed with special care to avoid false positive connections. Moreover, in the presence of pathological tissue, fiber reconstruction methods suffer from cerebrospinal fluid contamination due to the inflammation of neuronal tissue. Novel algorithms based on high angular resolution imaging are specifically designed to minimize the effects of this contamination but neglect substantial information (i.e., multi-shell diffusion data)[15]. In this regard, we designed a hybrid pipeline to use each method where it is best suited (Fig. S6) while, if available, using all the diffusion shells acquired. The SS3T method needs to be used only inside the tumor hence only discarding multiple diffusion shells in the lesion area. Further research

should try to assess whether independently running SS3T with each shell can be combined or not.

On the other hand, Anatomically Constrained Tractography (ACT) has been shown to increase the correspondence between generated streamlines and real fiber bundles, but its usage is not suited for tumoral regions. Automatic segmentation algorithms often delineate tumors with gray matter causing premature stopping of tracking algorithms[19]. To bypass this effect, Aerts and colleagues[5] artificially copied the segmented brain tissue from the contra-lateral hemisphere to the damaged region. However, this strategy neglected the fact that a brain tumor could have critically altered the white and gray matter structures invalidating the proposed copied segmentation (Fig. S6a). Alternatively, these constraints can be dropped at the price of losing anatomical correspondence between the tractogram and the white matter pathways[15].

In this direction, our pipeline used anatomical constraints outside tumors while entirely relying on the diffusion signal inside the tumor. Importantly, the relaxation of the anatomical constraints in peritumoral tissue together with the incorporation of intra-tumor FODs yielded well-known tracks that were otherwise truncated (Fig. 3b, c; see also Figs. S10–S13). Brain tumors are heterogeneous in size and location; thus, it is important to emphasize that our pipeline gradually reduces to a "standard" setting the smaller the tumor is. We studied the scale-free properties of the structural networks derived from our pipeline and observed that they asymptotically converged to the properties of networks derived without considering peritumoral fibers (Fig. S6d). For example, if a tumor were of small size and located mostly within gray matter, most of the peritumoral fibers would remain unaffected. However, in more critical cases, such as the ones shown here, the artificial truncation of streamlines derived from canonical approaches results in a larger percentage of disconnections than our hybrid pipeline (Fig. 3d, e). Crucially, network neuroscience studies might rely on these artificial disconnections to report structural disconnections without truly assessing whether they are indeed present or introduced[3,19]. Furthermore, the deletion of important fiber tracks could have detrimental effects on surgical planning given the evermore common use of tractography in pre-operative settings[41], needing personalized models that consider microstructural properties of the tumors for more accurate tracking alternatives[12].

Alternative solutions to fiber tracking inside brain tumors include multicompartment diffusion tensor models but given their impossibility to resolve complex white matter structures as well as using diffusion signal from a single shell, we chose not to use them[13,14].

As a last contribution, with the reconstructed structural networks, we train a machine learning model to predict connectivity rearrangement after brain surgery. Brain tumors display high heterogeneity including size, location, histology, grade, and infiltration in gray matter areas, among others. Consequently, networks of brain tumor patients also show great variability (Fig. 4). Furthermore, interindividual variability in terms of brain plasticity and network rearrangement after surgery represents another relevant source of complexity when trying to understand and predict the organization of brain networks[42].

To reduce the impact of both factors, previous work[43-46] guiding predictions in a different context with networks from healthy subjects achieved good results, even when considering simple methods[32]. As such, we designed a flexible anatomical prior that was used to filter unplausible connections. We framed the problem in the Bayesian domain, which permits this prior to be backpropagated during the training phase (see Supplementary Material). Then, highly plausible connections are naturally given more importance when minimizing the loss function while, at the same time, successfully discarding improvable edges.

Neural architecture design in deep learning is itself an exciting and constantly evolving field. Nonetheless, it is well known how the choice of a specific architecture introduces a bias[47]. A natural choice for predicting brain connectivity is graph neural networks. However, as stated in the introduction they are not exempt of problems[26]. To overcome them, we design a one-hidden layer non-linear regressor.

Fully connected layers have achieved great success in prediction studies of both clinical[48] and relational features[28] when adequately corrected for overfitting confounds. The model was trained with intercalated validation steps and its performance was evaluated using Leave One Out cross-validation. The generative model achieved lower reconstruction errors than the chosen benchmarks (Table 1; see also Fig. 4). We also trained the same model using brain networks obtained without the hybrid tracking pipeline (see Methods). Interestingly, results were slightly worse (Table S2; see also Table S3), suggesting that our hybrid method is equivalent if not better than current state-of-the-art fiber reconstruction pipelines.

Despite complying with normality assumptions, the reconstruction errors for two subjects were consistently low and high. An interesting paradigm in statistics emphasizes those data points that deviate from the trend. When looked from this perspective, the potential of deep learning models to uncover complex patterns translates into the increased ability to detect individual data points that deviate from, now more complex non-linear, statistical relationships.

The accuracy of the FCNET prediction was correlated with tumor size, achieving higher scores for small tumors (Fig. 5c, d). The chances of affecting long-distance structural connections naturally increase with the size of the tumor. Long-distance connections are expected to carry high metabolic costs[36]. Moreover, long-range connections in structural networks have been shown to emerge during the early stages of development in C. elegans[49], mice hippocampal circuits[50], and primates[51], although the debate is still active in the latter case. Therefore, directly reestablishing a long-range connection (i.e., end-to-end) to preserve rich-club and small-world topologies[5] might not be feasible. A possible way to circumvent these constraints would require several indirect connections leading to complex as well as non-normative patterns elusive to the model trained.

All but one patient suffered from grade I meningiomas, while all gliomas were of grades II and III. Interestingly, FCNET's prediction accuracy is less sensitive to tumor histology than grade. According to previous studies, tumor grade and histology have the greatest influence on survival rates in high-grade (III–IV) glioma patients[52]. Joint understanding of survival rates and connection rewiring is challenging, due to poorly established structure-function relationships. Given the results shown, however, we suggest that grade instead of histology is a better indicator of normative patterns of connection reestablishment. In addition, the last years have been characterized by increasing evidence on the mutational profile impact of survival rate of gliomas[53]. This means that gliomas can have completely different survival rates according to their molecular profile[53,54].

The location of the tumor did have an impact on the prediction score, although it was borderline significant. The predictions for frontal tumors showed slightly higher accuracies. Frontal areas are associated with higher metabolic activity[55] causing energy expenditure, shaping as well as increasing the costs of network rewiring[56]. On the other hand, frontal regions are also the endpoints of fiber bundles, mainly involving both short- and long-range connections. Consequently, these two factors might compensate for each other, allowing for rather normalized patterns of network rewiring easily captured by the prediction model. However, it is difficult to assert the validity of this result as frontal tumors represented slightly more than half of the total. Thus, FCNET might very well be overfitting to these patients although we are inclined to discard this hypothesis given the precautions taken into consideration (Fig. 5a; see also Methods).

In contrast to the functional results discussed previously, the accuracy was independent of the periventricular features. This is understandable since the ventricles contain cerebrospinal fluid and no biologically relevant streamlines could be reconstructed within the ventricular system.

Interestingly, despite not being trained on it, predictions from both the FCNET and the alternative model show essential topological properties found in brain networks. The predictive model generates weighted networks that, opposite to their simpler unweighted siblings, follow lognormal rather than scale-free distributions (Fig. 6; see also Supplementary Materials). This effect has been found in numerous studies on mice and macaques[57,58],

suggesting that rapid decay and high variance in connection strength arise from distance-dependent wiring costs[59].

The main limitation of this study is the small sample size and the lack of an external validation dataset. Heterogeneity could also impact the results. Despite not showing inter-group differences in the time interval between scans, individual trajectories should also be considered, and thus investigated in more detail. However, to overcome the heterogeneity present in the data, we conducted some independent analyses for several groups. As such, all statistical results should be interpreted with extreme caution. Nonetheless, small sample sizes are found very often in studies dealing with patients with brain tumors[1–5,41] possibly due to current standardized protocols disregarding pre-surgery functional acquisitions, rather simple diffusion sequences and the lack of multi-site initiatives[60–62]. Also, carefully designed metrics can pinpoint existing phenomena, as is often seen in medical case studies. Further work should also reveal the predictive potential of the score used in this work regarding cognitive impairment and perhaps local control and survival.

In terms of the machine learning model accuracy and reliability, due to a small sample size, we were limited as to which Deep Learning methods were usable. Recent progress in Geometric Deep Learning and Graph Representation Learning[24,25] have yielded very promising results which are already showing great potential in medical imaging applications. Furthermore, it has been proven that topological guidance of graph neural networks drastically increases accuracy in highly heterogeneous data[27]. However, all these methods require huge datasets which may not be available for medically sensitive problems such as the one studied here. Even more, the topology and structure of networks suffering from brain tumors are not yet clearly understood, introducing a new layer of complexity as to what measures should be used to guide the training. Nonetheless, further work should find an optimal compromise to exploit these useful features in smaller datasets.

To summarize, detailed analysis in the frequency domain revealed local and distributed abnormalities in resting-state time series and connections while establishing a potential causal link between them. We also proposed a pipeline for fiber tracking that used more diffusion signal than previous attempts, as well as anatomical constraints. Lastly, we used these structural networks, which included intra-tumor streamlines and connections, to train a machine learning model to predict and study structural brain connectomes after surgery. The model achieved competent accuracies, disclosed tumor-dependent plasticity patterns, and preserved biological topologies. In summary, our results showed that brain tumors are both functionally and structurally dynamic, strengthening the need for more targeted MRI and neuro-oncology studies.

## Methods
### Acquisition and usage of MRIs
A detailed explanation of the participants as well as the acquisition of the data is already available[63]; nonetheless, for the sake of transparency, we briefly present some crucial aspects. Subjects were asked to undergo MR scans both in pre- and post-surgery sessions. Out of the 36 subjects that agreed to take part in the pre-surgery session (11 healthy [58.6 ± 10.6 years], 14 meningioma [60.4 ± 12.3 years] and 11 glioma [47.5 ± 11.3 years]), 28 were scanned after surgery (10 healthy [59.6 ± 10.3 years], 12 meningioma [57.9 ± 11.0 years] and 7 glioma [50.7 ± 11.7 years]). The post-surgery scan session took place during the first medical consultation at the hospital after the surgical intervention (mean: 7.9 months, range: 5.2–10.7 months). There were no differences in the time intervals between the groups (meningioma [243 ± 12 days], glioma [223 ± 15 days], $p = 0.328$, two-tailed U-test). As a result, 19 pre- and post-surgery pairs of structural connectomes were usable as training and testing data. All brain tumors were classified as grade I, II, and III according to the World Health Organization. All ethical regulations relevant to human research participants were followed[63].

Each MR session consisted of a T1-MPRAGE anatomical scan (160 slices, $TR = 1750$ ms, $TE = 4.18$ ms, field of view = 256 mm, flip angle = 9°, voxel size $1 \times 1 \times 1$ mm³, acquisition time of 4:05 min) followed

by a multi-shell HARDI acquisition (60 slices, $TR = 8700$ ms, $TE = 110$ ms, field of view = 240 mm, voxel size $2.5 \times 2.5 \times 2.5$ mm$^3$, acquisition time of 15:14 min, 101–102 directions $b = 0, 700, 1200, 2800$ s/mm$^2$) together with two reversed phase-encoding $b = 0$ s/mm$^2$ blips to correct susceptibility-induced distortions[64]. Resting-state functional echo-planar imaging data were obtained (42 slices, $TR = 2100$ ms, $TE = 27$ ms, field of view = 192 mm, flip angle = 90°, voxel size $3 \times 3 \times 3$ mm$^3$, acquisition time of 6:24 min). The TR was accidentally changed to 2400 ms after 4 control subjects, 5 meningioma patients and 2 glioma patients were scanned changing the times of acquisition to 7:19 min. For all the subsequent Fourier analyses, this TR mismatch is solved by adding zero padding and truncating the shorter time series to ensure that equivalent spectrums were sampled by the Python methods (for further details see Supplementary Material).

Additionally, segmented lesions including the edema, non-enhancing, enhancing, and necrotic areas were available. Tumor masks were obtained with a combination of manual delineation, disconnectome[63], and the Unified Segmentation with Lesion toolbox[4]. To identify the tumor core of gliomas, two clinicians with more than thirty and ten years of experience performed and independently validated the segmentations using 3D slicer. Data only allowed for the identification of the tumor cores; hence we subtracted the resulting cores from the whole lesion to obtain a "non-necrotic" region for each of the patients diagnosed with a glioma-like tumor.

### Pre-processing of MRIs
High-resolution anatomical T1 weighted images were skull-stripped[65], corrected for bias field inhomogeneities[66], registered to MNI space[67], and segmented into 5 tissue-type images[68]. Diffusion-weighted images suffer from many artifacts all of which were appropriately corrected. Images were also skull-stripped[65], corrected for susceptibility-induced distortions[64], denoised[69], freed from Gibbs ringing artifacts[70] and corrected for eddy-currents and motion artifacts[71]. The preprocessed images were then co-registered to its corresponding anatomical template (already in MNI space)[67], resampled to a 1.5 mm$^3$ voxel size and eventually corrected for bias field inhomogeneities[66]. After motion correction as well as registration to the MNI template, the B-matrix was appropriately rotated[72].

Functional data was preprocessed with fMRIprep[73] and the eXtensible Connectivity Pipeline (XCP-D)[74] which are two BIDS-compatible apps that perform all recommended processing steps to correct for distortion artifacts in functional data. Regression of global signal has been shown to improve denoising in BOLD series without excessive loss of community structure[75]. In total, 36 nuisance regressors were selected from the nuisance confound matrices of fMRIPrep output which included six motion parameters, global signal, the mean white matter, the mean CSF signal with their temporal derivatives, and the quadratic expansion of six motion parameters, tissues signals and their temporal derivatives[76]. Volumes with framewise displacement higher than 0.3 mm were regressed out. Although smoothed time series were available, our analysis did not consider them. All specific steps were common to all subjects, both control and brain tumor patients. All images (T1s, T1 segmentations, diffusion, lesion masks and functional) were eventually co-registered to MNI space for comparison.

### Assessment of default mode network and tumoral BOLD Signals
BOLD signals belonging to the DMN were identified with the Gordon functional Parcellation[77]. More precisely, each one of the 41 regions classified as "Default" by the parcellation image was used as a binary mask to extract the time series from the functional image. For each subject (patient and control), the pair-wise Pearson correlation coefficient between time series was computed to obtain a functional connectivity matrix. The spatial overlap between DMNs and tumor masks was computed by summing all the voxels in the lesion mask belonging to one of these 41 regions. To normalize this score, we divided the resulting number by the number of voxels belonging to each one of the 41 regions labeled as "Default". Note that, with this definition, an overlap of 1 would mean the presence of a

tumor the size of the entire DMN.

$$\text{Overlap} = \frac{|Tumor \cap DMN|}{|DMN|} \quad (1)$$

Moreover, the spatial distance between the center of mass tumor and the DMN was computed by averaging the Euclidean distances to the center of mass of each one of the DMN nodes.

The DMN of the patients was compared to the mean of the healthy networks with two different metrics to assess (1) differences node-wise and (2) the Richness of the networks. Node similarity was assessed by computing the mean Pearson correlation between the same nodes in two different networks. For that, each row in the adjacency matrices was treated as a vector and compared with the same row of all matrices from the healthy subjects. After iterating through all nodes in the DMN, the mean and standard errors were computed for comparison. Furthermore, to assess the complexity of a given network, we computed the absolute difference between the distribution of correlations building the network and a uniform distribution[30]. We refer to this score as $\Theta$ Richness:

$$\Theta = 1 - \frac{m}{2(m-1)} \sum_{\mu=1}^{m} \left| P_\mu\left(r_{ij}\right) - \frac{1}{m} \right| \quad (2)$$

where $m = 15$ is the number of bins of the histogram estimating the distribution of correlations in the network $P_\mu(r_{ij})$. Zamora-López and colleagues showed the robustness of the quantity in Eq. (2) with regard to the value of the parameter $m$. However, sensible choices range from 10 to 20 to ensure a sufficiently rich approximation of $P_\mu(r_{ij})$. The changes in richness $\Delta\Theta$ across patients were obtained by computing the difference relative to the richness of the mean DMN obtained from control subjects: $\Delta\Theta = \Theta_{Patient} - \Theta_{Healthy}$.

A similar procedure was followed to study BOLD signals inside the lesioned tissue. For each patient, the binary mask containing the edema was used to extract the time series from the patient, as well as from all control subjects. Consequently, BOLD signals in lesioned regions of the brain were comparable to 11 healthy signals from the same region. No network was computable in this case, making the use of Eq. (2) pointless.

### Fourier analysis of BOLD signals
To compare time series between subjects, we computed the Real Fast Fourier Transform of the BOLD series. This allowed us to compare the power spectrum of two or more signals regardless of, for example, the dephasing between them. Let $A_\omega$ be the amplitude of the component with frequency $\omega$. Then, the total power of the signal can easily be obtained by summing the squared amplitudes of all the components:

$$P_T = \sum_{\forall \omega} |A_\omega|^2 \quad (3)$$

With the Fourier decomposition, we could also characterize the power distribution of the signals as a function of the frequency. Analogous to Eq. (3), we summed the squared amplitudes corresponding to frequencies inside a bin of amplitude $\Delta\omega$.

$$P_{\omega_c} = \frac{100}{P_T} \cdot \sum_{\forall \omega \in [\omega_c - \Delta\omega, \omega_c]} |A_\omega|^2 \quad (4)$$

Since each signal had a different $P_T$, to compare between subjects and/or regions, we divided the result by the total power $P_T$ and multiplied by 100 to make it a percentage. Arbitrarily, we chose the parameter $\Delta\omega$ for each subject so that each bin included 10% of the total power. The qualitative results did not depend on the exact choice of the bin width.

Similarly, we computed the cumulative power distribution $CP_\omega$ by summing all the squared amplitude coefficients up to a certain threshold. For consistency, we measured the $CP_\omega$ as a percentage score

and chose the thresholds to be multiples of exact percentages i.e., $\omega' \propto 10\%, 20\%, \ldots$).

$$CP_{\omega_c} = \frac{100}{P_T} \cdot \sum_{\forall \omega \in [0, \omega_c]} |A_\omega|^2 \qquad (5)$$

Both the power distribution $P_\omega$ and cumulative power distribution $CP_\omega$ can be used to compare dynamics between time series, but they have the inconvenience of not being scalar numbers. Furthermore, computing any distance-like metric (i.e., KL divergence) between these distributions across subjects would not yield any information of whether BOLD signals had slower dynamics (more power located in low frequencies) or the opposite (i.e., DMN in healthy and patient).

To overcome this, we designed a DAS between time series based on the difference between two cumulative power distributions. It is worth noting that in the limit $\Delta\omega \rightarrow 0$, the summations in Eqs. (2), (3), and (4) become integrals simplifying the following mathematical expressions. The DAS between two BOLD signals $i, j$ was computed as the area between the two cumulative power distributions:

$$DAS(i, j) = \int d\omega \left( CP_\omega^i - CP_\omega^j \right) = -DAS(j, i) \qquad (6)$$

Finding a positive $DAS(i, j)$ would mean that time series $i$ had slower dynamics than time series $j$ since more power is accumulated in lower frequencies with respect to the total. Throughout this manuscript, DASs were defined as the difference in power distribution between patients and the healthy cohort. For a simplified and, hopefully comprehensive, example, we kindly refer the reader to Fig. S1. To characterize a specific DMN, all these measures were computed for each region separately and then averaged [mean ± SEM]. As opposed to the $\Theta$ Richness, the DAS was computable both for DMNs and tumors since it only required two temporal series rather than a complete distribution. To compute absolute values of this score, the DAS for each region (or tumor) was made strictly positive. Only then average between regions and subjects was performed. Notably, these two operations are not interchangeable.

For the score defined in Eq. (6) to make sense, the Real Fast Fourier Transform of the time series needed to be computed using the same frequency intervals, which, in short, implied that the time duration of the signals needed to be equal. For functional images with different $TRs$, this was solved by adding zero-padding to the shortest signal to match the same time duration (Fig. S14). Further permutation analyses on a reduced subset of subjects with identical TRs confirmed the tendencies reported in the text (Fig. S15).

### Reconstruction of tumoral structural brain networks

To ensure a detailed subject-specific network, we used a state-of-the-art pipeline to obtain brain graphs while at the same time not neglecting tracts inside lesioned regions of the brain (i.e., brain tumors). We combined two reconstruction methods, yielding two different tractograms and three connectivity matrices. Roughly, the first tractogram aims at reconstructing white matter fibers using non-contaminated diffusion signal, while the second one carefully assesses the presence of meaningful diffusion signal in perilesional and lesioned tissue. Later, for each tractogram, a personalized connectivity matrix can be obtained and combined to yield a unique abstraction of the brain in surgical contexts. A schematic workflow of the pipeline is in Fig. 3a, and a detailed account of the parameters is in Table 2.

The first branch of the method consisted of a well-validated set of steps to reconstruct the network without considering lesioned regions of the brain. To ensure this was the case, we used a binary brain mask that did not include the segmented lesion (i.e., we subtracted the lesion from the brain binary mask). This step was added for consistency with the logic of not tracking within the lesion. Nonetheless, the steps were repeated without this mask and the results were found to be almost identical (Fig. S6). This was expected as multi-shell methods highly disregard cerebrospinal fluid

contamination inside the lesion[15]. The lesion mask was added to the 5 tissue-type image to be considered as pathological tissue[78]. Within this mask, for each b-value shell and tissue type (white matter, gray matter, and cerebrospinal fluid) a response function was estimated[79]; and the fiber orientation distribution functions (FODs) were built and intensity normalized using a multi-shell multi-tissue (MSMT) constrained spherical deconvolution approach[80]. Within the same binary mask excluding potentially damaged tissue, anatomically constrained whole-brain probabilistic tractography was performed using dynamic seeding, backtracking, and the iFOD2 algorithm[68,81]. The total number of streamlines was set to 8 million minus the number of streamlines intersecting the lesion (see below). We used spherical-deconvolution informed filtering to assign a weight to each generated streamline and assess their contribution to the initial diffusion image[82]. Finally, a *healthy* structural connectivity matrix was constructed by counting the number of *weighted* streamlines between each pair of regions of interest as delineated by the third version of the Automated Anatomical Label atlas[83].

Next, to consider fiber bundles that originate and traverse lesioned tissue, a recent method for reconstruction was used only in the segmented lesion[17]. The coined Single-Shell-3-Tissue Constrained Spherical Deconvolution (SS3T) algorithm uses only one diffusion shell and the unweighted $b = 0$ volumes. We used the shell with the highest gradient strength (i.e., $b = 2800$ s/mm$^2$) as it offered the best white-matter contrast[15,80]. These FODs were reconstructed, and intensity normalized only inside the lesion mask using the same underlying response function as estimated earlier in the healthy tissue. We merged the reconstructed FODs with the previously obtained with the multi-shell algorithm (Fig. 3a CENTER). It is important to note that both images were in NIFTI format, co-registered, and non-overlapping, therefore making this step straightforward. Anatomical constraints were no longer suited since white- and gray-matter are compromised inside the lesion and in the perilesional tissue. Even more, regardless of the FOD reconstruction procedure, the anatomical constraints caused fibers to stop around the edema since those surrounding voxels were (nearly-)always segmented as gray matter (see Fig. S6). We used dynamic seeding only within the masked lesion and whole-brain probabilistic tractography with backtracking to estimate white-matter fibers within the whole-brain mask[68,81]. The number of streamlines was set as the average number of streamlines intersecting the lesion in the healthy cohort. We superimposed the lesion on the tractograms of each control subject and tallied the overlapping streamlines[78]. This was important given that each lesion was in a different location and the natural density of streamlines in that specific location differed. This subject-specific streamline count controlled that the tract densities were comparable to the healthy cases (Fig. 3b–e; see also Figs. S10–S13). Spherical-deconvolution informed filtering[82] was applied to ensure that each streamline adequately contributed to the *lesioned* diffusion signal (i.e., filtering was applied inside the lesion mask). Then, a *lesion* structural connectivity matrix was constructed similarly to the previous case.

$$N_{streamlines\,in\,lesion} = \frac{1}{N_{control}} \sum_{i=1}^{N_{control}} \sum_{streamline=1}^{streamlines} 1 \begin{cases} 1 \text{ if } streamline \in \text{Lesion} \\ 0 \text{ otherwise} \end{cases}$$
$$(7)$$

Finally, we merged the two available connectivity matrices to reconstruct a lesioned structural brain network. To do so, we employed a greedy approach where we took the maximum connectivity strength for each pair of regions:

$$\omega_{ij} = \max\left( \omega_{ij}^{healthy}, \omega_{ij}^{lesion} \right) \qquad (8)$$

Thus, for each pre-operative scan, a total of 3 different connectivity matrices were available: the healthy connections, the (potentially) lesioned connections, and the full lesioned structural network. The networks from

## Table 2 | Tractography parameters

| Tractography parameter | Value |
| --- | --- |
| Maximum SH coefficient order (*lmax*) | 8 |
| Number of seeding points | 80 – million |
| FOD amplitude cutoff | 0.08 |
| Min/max streamline length | 3/280 – voxel size units |
| Euler integration step size | 0.5 – voxel size units |
| Maximum angle | 45 – degrees |
| SIFT2 Tikhonov regularization | 0.1 |
| SIFT2 Asymmetric Total variance regularization | 0.08 |
| Total number of streamlines | 8 – million |

A detailed summary of the parameters and values used for all fiber orientation distribution functions, tractography, and regularization steps. Note how the total number of streamlines is constant (i.e., 8 million) for all subjects but each subject has a unique number of streamlines being generated within the lesioned tissue thus altering the number of streamlines outside the lesion.

the control subjects and post-operative scans from patients were reconstructed using a usual multi-shell multi-tissue pipeline without the binary lesion-free mask but with the same parameters (see Table 2). Note that the 3rd version of the Automated Anatomically Labeled Atlas has 4 empty regions out of 170 to maximize compatibility with the previous versions.

### Anatomical priors and Bayesian filtration of predictions

As suggested by previous works, guiding learning with healthy cohorts should be useful to inform predictions[43–45]. Brain graphs are notoriously heterogeneous when considering age-related differences. To take this into account, we selected subjects with significant age overlap between healthy subjects and patients in both tumor types. However, we did not consider sex segregation, since structural differences are rather unclear[84]; even more, the sample size for each subgroup would be severely reduced. We built a prior probability distribution of healthy links to guide the predictions using a *thresholded* average of the set of connections present in this healthy cohort (see Supplementary Material). This thresholded average allowed us to control for the inclusion (or exclusion) of spurious connections, while minimizing the false positive rate of connections[85].

### Training and testing the machine learning model

For each reconstructed network, a total of 13695 normalized edges needed to be reconstructed, thus making the problem ill-posed. Nonetheless, as argued in the introduction, we hypothesized that a fully connected network adequately guided with anatomical information could capture essential properties (see Supplementary Material). We evaluated the model using Leave One Out Cross Validation, therefore, training and testing a total of 19 models or 19 *folds*.

The high number of reconstructed fibers yielded high values for the connectivity between ROIs ($\sim10^3$). To prevent numerical overflow as well as to enhance differences in lower connections, all weights $w$ were normalized by computing $\log(1 + \omega)$ before feeding them into the artificial deep neural network.

The model consisted of a 1 hidden layer deep neural network which was trained minimizing the Mean Squared Error (MSE) between the output and the ground truth determined from the MRIs (see Supplementary Material). The weights were optimized using stochastic gradient descent with a learning rate of 0.01 and 100 epochs to avoid overfitting. Evaluation metrics included the Mean Absolute Error (MAE), Pearson Correlation Coefficient (PCC) and the Cosine Similarity (CS) between the flattened predicted and ground truth graphs. The topology of the generated networks was evaluated by computing the Kullback-Leiber and Jensen-Shannon divergences between the weight probability distributions of the generated and real graphs.

Leave One Out cross-validation was done using 18 connectomes to train each one of the 19 models. For each model, the training data was randomly split into train (80%) and validation (20%) sets to prevent overfitting. Validation steps were run every 20 training epochs. For each fold, the testing of each model was done in the *left-out* connectome (Table S1).

### Statistical and topological network analysis

Statistical tests and *p*-value computations were done with Scipy's stats module and in-house permutation scripts. Unless stated otherwise, we used one-tailed hypotheses only when addressing the significance of strictly positive magnitudes combined with non-parametric methods. Non-negative magnitudes cannot be tested for negative results and do not need to satisfy normality.

The Leave One Out cross-validation approach yielded a pool of 19 subjects that were independently tested. For each metric, we computed the z-score by subtracting the mean and dividing by the standard deviation of the sample. Despite verifying that all of them were normally distributed, we ran parametric and non-parametric statistical tests due to the small sample size. Topological metrics were computed using the Networkx Python library[86]. Since the brain graphs were weighted, we computed a weight probability distribution instead of the more common degree distribution (see Supplementary Material). To compare the weight probability distributions of two graphs, we computed the Kullback-Leiber as well as the Jensen-Shannon divergences. The Jensen-Shannon divergence has the advantage of being both symmetric and normalized between 0 and 1 therefore interpretable as a distance between two distributions (i.e., predicted vs ground truth).

### Reporting summary

Further information on research design is available in the Nature Portfolio Reporting Summary linked to this article.

### Data availability

The original data is publicly available at OpenNeuro[63]. The processed data necessary to reproduce this study is also publicly available[87].

### Code availability

For the processing of MRIs several software packages were combined and integrated in a flexible Python 3.9.7 pipeline[73,74,78,88–90]. For the machine learning model, PyTorch and CUDA libraries were used. The code is publicly available[87].

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

## Acknowledgements

We thank Ricardo Budai and Rosmary Blanco for the segmentations of the tumor nodules and feedback. The publication was created within the project of the Minister of Science and Higher Education "Support for the activity of Centers of Excellence established in Poland under Horizon 2020" on the basis of the contract number MEiN/2023/DIR/3796. This project has received funding from the European Union's Horizon 2020 research and innovation program under grant agreement No 857533. This publication is supported by Sano project carried out within the International Research Agendas program of the Foundation for Polish Science, co-financed by the European Union under the European Regional Development Fund. This research was supported in part by the PLGrid infrastructure. Computations have been partially performed on the ARES supercomputer at ACC Cyfronet AGH.

## Author contributions

J.F.-R. and A.Cr. conceptualized the study; J.F.-R. analyzed the data; J.F.-R., A.Ca., F.S. and A.Cr. wrote the paper.

## Competing interests

The authors declare no competing interests.
