## [Peer review file · Communications Biology]

Reviewers' comments:

Reviewer #1 (Remarks to the Author):

This research showed a prediction of structural rewiring post-surgery and shed some light on the impact of oedema in functional and structural signals. The topic is interesting, and the results are well described. However, there are some mistakes. I suggest a major revision is required for the manuscript.

1. The article was rich in content, but maybe lack a clear logical thread. Could the author articulate the logic of the main work in a few sentences?
2. The most important issue is the too-small sample size and lack of independent validation. Currently, multicenter and large sample studies are the trend of development, and the small sample size and lack of independent validation set in this study is a fatal flaw. The representativeness of the sample is questionable, and there is a possibility of overfitting.
3. In line 661, please provide a description of the time interval between two scans of the three groups.
4. In line 679, some fMRI data had inconsistent parameters between the two scans, which may cause additional effects.
5. In line 726, how the two different metrics of the healthy networks were calculated ?
6. In line 888, why used one-tailed tests in the most statistical analysis in this paper?
7. In figure 1B, how the "relative PT" was calculated ?
8. Why some correlation analyses used absolute values and some did not?

Reviewer #2 (Remarks to the Author):

COMMSBIO-22-4195

The authors present a carefully conducted study of tumor impact on the brain in relation to surgical intervention using structural and functional MRI data and machine learning methods. The authors demonstrate another approach to measure the impact of tumor and neurosurgery on brain function and structure.

The manuscript is carefully written. However, due to the use numerous methods, it is very long and addresses many aspects which makes it difficult to read.

Major

1. The authors claim that a new method for fiber tracking pipeline can use anatomical information and still reconstruct fiber bundles in tumoral and peritumoral tissue. However, this is only weakly clarified and illustrated in the manuscript.
2. The authors included a low number of tumor patients. The results are interesting regarding tumor patients. However, how do the authors justify the study design of a mixture with a completely different tumor entity, meningiomas, which do not grow from glial cells, and in which the influence of the tumors on white matter is different? Furthermore, gliomas 1-4 also differ significantly, be it in terms of kinds, aggressiveness, or growth potential.
3. The authors claim that "The resulting brain networks captured streamlines and connections which were previously non-existent when using each algorithm alone (Fig. S6A)." A) I strongly doubt that msmt won't reveal FOD's within the tumor region. In Fig. 3, the authors show a brain mask which excludes the tumor (regarding their use of msmt). If they passed a 5TT image without a lesion mask, then they would obtain also FOD's in the voxels of the tumor, cf. my other comment

in the Minor section. B) Why do the authors merge to different algorithms to estimate response functions? I suggest using one, e.g. MSMT_5TT or dhollander or then SS3T to have the same underlying response function in the resulting tractogram. C) Please also explain the rationale for the usage of SS3T via one extracted shell of the multi-shell data since it is designed for single shell data, while all the shells would be available. D) Performing tractography exclusively within the tumor region is controversial. How are these streamlines controlled? I propose to perform tractography with FOD datasets that are not constrained by a tumor mask or combined and to carefully examine the FOD's in the tumoral region so that signal estimation considers the whole brain and only one response function is used.

4. The authors use two different atlases to obtain functional (Gordon parcellation) and structural (AAL) connectomes. Have they considered to use another atlas that would fit both acquisition schemes?

5. I miss in the abstract a short info about the result that the frequency domain showed local and distributed anomalies in time series.

Minor

1. The authors do not provide the degrees of freedom regarding the correlation test results. Please provide all information.

2. Unfortunately, the authors also only provide p-values for their ANOVA and do not provide the entire picture. The authors need to provide the degrees of freedom (between groups, within groups) and the F value, cf. also t-tests.

3. There is a newer, improved version, for tractogram filtering, SIFT 2, as the authors are certainly aware of. Why did the authors use the old SIFT version?

4. What is the rationale for selecting 4M streamlines with an atlas consisting of 166 nodes? Have the authors tested reproducibility with sifted 4M streamlines and 166 nodes? I suspect that the number of streamlines needs to be increased to get reproducible tractograms.

5. The authors combine the FOD's obtained from MSMT with a tumor mask and SS3T within the tumor mask. How different is the MSMT signal from MSMT without a tumor mask? MSMT is stable, especially with the multi-shell dMRI acquisition and will also provide FOD's in the peritumoral area.

6. There is a typo on Line 283 “.” .

7. Figure S6, A) How can the authors prove that these streamlines are not erroneous? B) In Figure S6 part A the streamlines in the tumoral area appear much denser, as if there were much more streamlines as in the other WM areas, please explain. C) The yellow tumor mask obscures the streamlines. Please provide an image showing the coronal and sagittal aspects as well, preferably with a transparent tumor mask and in relation to the FOD's. D) I couldn't find a description for Figure S6 part C – a modified 5TT image?

8. MSMT and SS3T show different crops in Figure 3.

9. Which activation function was used for the hidden layer and the output layer? The model architecture is described sparsely.

10. Please clarify the pathologies of the patients used in Fig. 4.

11. I would prefer to see more visual results regarding the obtained FOD's in the tumoral areas and a comparison to the use of MSMT response function in the tumoral area to visually inspect differences to the results from SS3T. Without this, the complicated additional effort with the combination of another response function is presented to the readership in a way that is insufficiently transparent.

Reviewer #3 (Remarks to the Author):

In this study the authors investigated the impact of brain tumors on functional and diffusion MRI. They found that intra-tumor fMRI signals were correlated with alterations in healthy tissue and resting-state networks. From a structural view, they designed a hybrid tractography pipeline capable of combining anatomical constraints and diffusion signals inside tumors, and further proposed a machine learning method to predict postsurgical structural rewiring. In general, the topic is interesting, and the findings have important clinical implications for tumor surgery. However, I still have some comments as follows:

1. The major concern I have with this paper is the comparison of BOLD signals. The experimental group used segmentation masks covering all parts of the tumor, from necrotic tissue to oedema, and their BOLD signals were compared with the signals in the same regions of the control subjects. However, a tumor may consist of different tissues, i.e., edema and necrosis, and therefore exhibits different functional signals. Similarly, a region with a mixture of neurons and tumor cells may show different functional characteristics. The authors should explain the rationale of directly comparing the signals in the whole tumors with that in the normal tissues.
2. There are many subjective perspectives in this paper that should be removed. For example, in line 110, "Fast and naïve inspection of those signals did not yield visual differences between tumoral and healthy signals"; in line 153, "visual differences between functional matrices (i.e., connectivity strengths) seemed to be anticorrelated with the magnitude of the DAS". The authors should focus on the findings supported by their data.
3. In line 113, the authors should clarify whether the DMN signals are from the same patient or from a healthy control.
4. In the first part of the results, please explain why the signals inside tumors are compared to the signals in the DMN but not in other regions?
5. In Table 1, please explain what the bold numbers represent?
6. In Fig. 5B left, the r values shown in the figure are inconsistent with the text, please explain.
7. In line 375, why tumors < 12.95 cm³ are defined as small tumors, and tumors larger than that are large tumors? Is there any reference for this criterion?

Reviewer #1

This research showed a prediction of structural rewiring post-surgery and shed some light on the impact of oedema in functional and structural signals. The topic is interesting, and the results are well described. However, there are some mistakes. I suggest a major revision is required for the manuscript.

(A) We appreciate the positive comments on our work. On the other hand, we also acknowledge the concerns, and we have tried to address them appropriately. **We have now provided our answers (A) in red and changes (C) in blue both in this document and in the revised manuscript.**

1. The article was rich in content, but maybe lack a clear logical thread. Could the author articulate the logic of the main work in a few sentences?

(A) We acknowledge that a summarizing statement was lacking at the end of the *Introduction* as well as at the beginning of the *Discussion* sections. As such, we have expanded those paragraphs in the hope that they improve the clarity of the written content.

(C) The last paragraph of the *Introduction* has been extended to: “In summary, we present several contributions in both functional and diffusion MRI domains in the presence of a brain tumor. Initially, we study BOLD signals within the tumor and how they relate to abnormal patterns of brain-wide resting-state connectivity and complexity. Secondly, we present our attempt to consider fiber tracking within the lesion combined with whole-brain tractography. Lastly, we study how the fibers in the lesion, as well as surgery, impact the structural connectivity after tumor resection.”.

The first paragraph of the *Discussion* has been extended with: “[...] Importantly, unlike previous studies, we focus on the signal present inside the lesion rather and how it explains connectivity reorganization in both modalities. [...]”.

2. The most important issue is the too-small sample size and lack of independent validation. Currently, multicenter and large sample studies are the trend of development, and the small sample size and lack of independent validation set in this study is a fatal flaw. The representativeness of the sample is questionable, and there is a possibility of overfitting.

(A) We share the reviewer’s concerns although unfortunately, our possible actions regarding this issue were rather limited. Nonetheless, we dedicated substantial efforts to try to resolve them both conceptually and practically.

Firstly, we would like to point out a substantial amount of valuable and recent research being done with similar sample sizes [1-7]. As already mentioned in the manuscript, the data was thoroughly described recently [3] and divided amongst similar groups in both quantitative analyses [1,2]. The main issue was the lack of publicly available data. There are large multicenter consortiums that provide free access to data, but they only consist of anatomical [8] or structural acquisitions [9].

Crucially, HARDI multi-shell diffusion MRI and functional MRI acquisitions are not standardized protocols according to recent guidelines from the National Cancer Institute [10]

and the European Association for Neuro-Oncology [11,12]. Clearly, priority is given to clinical and surgical utility, ultimately leading to an increase in the patient's well-being. Postoperative MRI acquisitions are intended to assess the extent of resection and disease progression, both commonly measured-enhanced anatomical MRI, although recent evidence suggests the use of PET MRI [13].

After extensive research, we could not find any data with pre- and post-surgery diffusion acquisitions. We also did not successfully find any usable data for pre-surgery fMRI together with control groups. The lack of recommendations to acquire post-operative diffusion scanners can be seen as the main cause of the absence of available datasets to conduct reproducible tractography studies such as the one we propose in this manuscript. Furthermore, functional MRI acquisitions are not included in standard pre-surgical routines further complicating access to them. Data on brain tumors are not often publicly released, possibly due to ethical compromises and sensitivity with the patients and their caregivers.

Regarding overfitting, we believe that we took all the necessary precautions recommended, namely train-validation split and cross-validation.

(C) Having acknowledged these concerns, we reiterate them in the *Limitations and Future Directions*. We have added references to previous studies (already present in the Introduction section). We have also added the three guidelines regarding acquisition protocols in the same section hoping to provide an overview of the landscape regarding MRI studies in brain tumor settings. The sentences read as follows: “[...] Nonetheless, small sample sizes are found very often in studies dealing with patients with brain tumors (Stoecklein, et al. 2020; Nenning, et al. 2020; Silvestri, et al. 2022; Aerts, et al. 2018; Aerts, et al. 2020, Sullivan, et al. 2023) possibly due to current standardized protocols disregarding pre-surgery functional acquisitions, rather simple diffusion sequences and the lack of multi-site initiatives (Ellingson, et al. 2015; Weller, et al. 2017; Weller, et al. 2021). Also, carefully designed metrics can also pinpoint existing phenomena, as is often seen in medical case studies. [...]”

3. In line 661, please provide a description of the time interval between two scans of the three groups.

(A) We have added more information regarding the time interval between acquisitions. We now mention that the time was determined solely by clinical practice. We have deleted the previous temporal information from the sentence and instead added a new one specifying in detail the time intervals very similarly as in the original publication [1].

(C) We have simplified this sentence “[...] 28 were scanned after a period spanning from 6 to 12 months in the post-surgery session [...]” to “28 were scanned after surgery [...]”. Instead, we have added the sentence: “The post-surgery scan session took place during the first medical consultation at the hospital after the surgical intervention (mean: 7.9 months, range: 5.2-10.7 months).”.

4. In line 679, some fMRI data had inconsistent parameters between the two scans, which may cause additional effects.

(A) Although unfortunate, it is true that this inconsistency might have introduced artifacts. We took inspiration from the original publications [1-3] despite the difference in the type of

analyses we performed. We carefully considered the implications of different repetition times (TRs) and their potential effect on our methods. Given that no specific concern was raised in this comment but rather a general statement, we try to give an overview of our reasoning and argue why we believe the actions we took suffice in reducing the risk of additional artifacts introduced by this inconsistency.

Briefly, we proposed to solve the TR inconsistency by padding the shorter time series to ensure a consistent frequency spectrum and truncating the resulting high frequencies of the extended signal. In doing so, we realized that in the original unrevised implementation of the code, there was a mistake in the last step of this process (i.e., the truncation). As a result, several statistics were incorrectly obtained. After correcting the mistake, the qualitative results did not change and, importantly, increased in significance. We comment in detail on the changes when answering comments 6 and 8. The updated figure is also attached at the end of this rebuttal.

(C) The last sentence of the first paragraph of the same subsection has been extended to describe in more concise terms how we solved the TR inconsistency. We have written it as follows: “For all the subsequent Fourier analyses, this TR mismatch is solved by zero padding and truncating the shorter time series to ensure that equivalent spectrums were sampled by the Python methods (for further details see Supplementary Material).”

To further clarify this issue, in the supplementary material we have added a whole new section and a new figure (Fig. S9 – find the code in the repository). The new section, located at the beginning of the supplements, is titled: *Padding and truncating Fourier transforms to solve the TR inconsistency between fMRI scans*. In it, we discuss the effects that different TRs might have on the results based on the existing literature [14-16]. In addition, we provide a comprehensive example of the exact procedure that we have followed to mitigate this unfortunate mistake made by the MR technician. We also attach a copy of the added figure (in lower resolution) and the caption below. A specific mention of this supplementary section was added in the Methods section below Eq. (6).

5. In line 726, how the two different metrics of the healthy networks were calculated

(A) We apologize for the lack of clarity of this sentence. The first metric we used was the Pearson correlation between the same nodes of two patients and then averaging over the nodes. The second metric was the θ Richness already described in the Methods.

(C) To improve the clarity, we rewrote the mentioned lines in the following way: “The DMN of the patients was compared to the mean of the healthy networks with two different metrics to assess 1) differences node-wise and 2) the Richness of the networks. Node similarity was assessed by computing the mean Pearson correlation between the same nodes in two different networks. For that, each row in the adjacency matrices was treated as a vector and compared with the same row of all matrices from the healthy subjects. After iterating through all nodes in the DMN, the mean and standard errors were computed for comparison. Furthermore, [...]”

6. In line 888, why used one-tailed tests in the most statistical analysis in this paper?

(A) We share the reviewer’s concern regarding the nature of one-tailed hypotheses. However, in some cases, the use of two-tailed tests is incorrect due to the asymmetry of the

tested magnitude. For example, a test dealing with an increase (with respect to none) in the absolute value of a result does not include negative results (i.e., the absolute value of the difference in power is strictly non-negative). In these scenarios, we believe that the use of one-tailed hypotheses is justified. However, in agreement with the raised issue, we have reduced the number of one-tailed hypotheses only to the cases where the above answer is applied. Importantly, we have incorporated a non-parametric test to support statements of significance or non-significance throughout the results section.

Notably, when testing the effect of the intersection between tumor and ventricles, there we had a clear prior belief that the signal might have been contaminated with cerebrospinal fluid (CSF) impacting the dynamics in a particular way: slowing the BOLD signal. Because of that, we decided to keep the one-tailed hypothesis. After the corrections applied in comment 4, the dependency remained non-significant but strong, hence was left unmodified (see changes below).

(C) In line 888 of the unrevised version of the manuscript, we have rewritten the details on hypotheses testing as follows: “Unless stated otherwise, we used one-tailed hypotheses only when addressing the significance of strictly positive magnitudes combined with non-parametric methods. Non-negative magnitudes can’t be tested for negative results and needn’t satisfy normality.”.

All the changes are specified in the modified manuscript, as they would have unnecessarily lengthened this rebuttal. To further increase transparency, we also modified the periventricular testing hypothesis here and in the manuscript.

- Line 235 has been rewritten to make our prior belief explicit. “If the tumor intersected the ventricles, we expected the cerebrospinal fluid to penetrate the lesion thus severely disturbing and slowing the signal recovered. In fact, tumors intersecting the ventricles showed a large positive DAS, although non-significant (Fig. 2E TOP; $p=0.31$, one-tailed U-test).”.

7. In figure 1B, how the "relative PT" was calculated ?

(A) We agree with the reviewer that further clarification was needed. We specified that the *relative* component of the magnitude refers to the default mode network (DMN) of the healthy subjects in the dataset.

(C) The captions of figures 1, S2, S3, S4, and S5 have been changed accordingly. We have added the following: The relative power was obtained by averaging over subjects and DMN regions and taking healthy subjects as the baseline (i.e., equal to 1).

Moreover, in the same subsection mentioned in comment 4, we have specified how the TR inconsistency was solved: “Noteworthy, the total power of the signal after the extension is not kept constant, but the distribution of the power is correctly maintained (Fig. S9). [...] However, all the absolute power analyses conducted in this study were done considering the unpadded signals – although the statistical results were identical to those obtained with the padded and truncated signals (i.e., up to the third decimal point).”.

The figures showing the results patient-wise slightly changed due to the power computation and the correction specified in comment number 4. As a result, Figs. 1, S2, S3, S4 and, S5 were redone. However, both qualitative and quantitative trends remained unaltered, hence not attached to this rebuttal.

8. Why some correlation analyses used absolute values and some did not?

(A) We used absolute values for some correlation analyses to study whether any kind of alteration in the dynamics could explain network abnormalities. Finding slower or faster oscillations might have similar effects on the network (such as in the similarity – Fig. 2A) or opposite effects (such as in the complexity – Fig. 2A RIGHT). A somewhat similar result was found in the case of spatial distribution of the tumors.

However, we acknowledge that transparency and clarity could be improved in this regard. For this purpose, we have added an additional analysis in the case of network similarity studying both the absolute value of the scores as well as the score without making it positive. Please, find below the resulting changes.

(C) It is worth noting that computing absolute values and then averaging over them is not equivalent to the reverse order (e.g., the mean of the absolute value is not equivalent to the absolute value of the mean). To avoid further confusion, we added two sentences in the methods, below Eq. (6) to make this difference explicit: “To compute the absolute values of this score, the DAS for each region (or tumor) was made strictly positive. Only then average between regions and subjects was performed. Notably, these two operations are not interchangeable.”.

We added the following sentence to explain the relationship between altered dynamics and network (dis)similarity. Both positive and negative scores contributed to the decrease in network similarity between healthy and patient. In the section *Temporal Dynamics and ...* of the results we have added: “We studied the similarity as a function of the direction in the alteration expecting it would decrease regardless of the change in dynamics for large scores. Indeed, we found a clear inverted “U-shaped” pattern that agreed with the interpretation that altered dynamics is a contributing factor in network reorganization (Fig. 2A Inset). Negative DAS was positively correlated with similarity ($r > 0$, $p = 0.006$, $df = 11$, one-sided permutation test) while positive DAS exhibited the opposite trend ($r < 0$, $p = 0.083$, $df = 14$, one-sided permutation test).”. The one-tailed hypotheses were important in this case because the null was a large value, NOT affecting the organization.

The linear fit between absolute DAS and absolute change in complexity has been added in Fig. 2A RIGHT.

In Fig. 2B LEFT we have explicitly added the information that alterations in the dynamics were uncorrelated with the distance between lesion and DMN. However, in the first subsection of the discussion, we have added a sentence further describing this relationship: “Interestingly, dynamic alterations in DMN regions increased rather linearly with the distance to the tumor, from faster oscillations in nearby tumors to slower ones in spatially separated lesions ($r = 0.442$, $p = 0.027$, two-tailed exact test). However, distance does not account for the difference in network rearrangements given that dissimilarity increases for both slower and faster dynamics.”.

In Fig. 2B RIGHT we now plot the correlation between DAS in the tumor and DMN without the absolute values. The conclusions remain unaltered. The relationship is stronger than previously and well-adjusted to the data. We have included the 95% confidence intervals (shaded regions) to increase the reliability of the trend that we report.

Both Figure 2 and the corresponding caption have been adequately updated to describe the changes. It can be found at the end of this rebuttal, below the modified figure.

At the end of the subsection *Temporal Dynamics and ...* (in the results) we have rewritten the last summary paragraph to include new information: “In conclusion, [...]. Slower signals increased the complexity of the patients’ network while faster oscillations decreased it. Crucially, the appearance of any sort of disturbance in the BOLD dynamics was associated with a poorer similarity with the DMN from healthy subjects. In the following section, we explore how and where these alterations may arise in brain tumor patients.”.

Updated figures

Figure S9 Solving the TR inconsistency using Fourier analysis. **A)** Two series sampled with different rates (TR=2400ms blue, TR=2100ms orange) while keeping the number of points constant at 180 points. As a result, the short TR series is shorter in time. The underlying signal consists of a combination of two sinusoidal signals, each one with different frequencies. Although the signal is the same, the sampled series differ slightly. Importantly, the TR mismatch in the real data is the same. **B)** The discrete Fourier sampling frequencies for the original series in A and the series after adding zero-padding. Note how the black and red hyphenated lines coincide after the truncation (frequencies to the right of the gray vertical line are discarded). **C)** Power spectrum and cumulative power of the two series before (LEFT) and after (RIGHT) zero-padding. Note that the cumulative power in the short unpadded series reaches 100% before the long series. This results in artificially altered DAS. After padding and truncation, these effects are successfully corrected.

Figure 2 Linking DMN and Oedema BOLD Signals. Pearson Correlations and the corresponding p-values are shown. Error bars correspond to the mean \pm SEM. **A) LEFT**, Correlations between node similarity as measured by the Pearson correlation coefficient and the Dynamics Alteration Score (DAS). The inset shows how alterations in the dynamics shape the organization of the network regardless of the direction of the displacement of the frequency spectrum. **RIGHT**, Alterations in dynamics correlate with alterations in network complexity. Change in the complexity of the patients' DMN with respect to the healthy pool (Directionality $\Delta\theta$ and Magnitude $|\Delta\theta|$) as a function of the DAS. **B) LEFT**, Scatter plot showing the null correlation of DAS with distance and overlap between lesion and DMN centroids. **RIGHT**, A strong linear trend was found between alterations inside the patients' tumor and alterations in the DMN as measured by the DAS. The orange line corresponds to the significant linear fit (two-sided exact test) and the shaded areas mark the 95% confidence interval. **C)** Differences in total power relative to healthy subjects (LEFT) and dynamics (RIGHT). Only the absolute

values were significantly different from zero (one-side t-test and U-test). **D**) Scatter plot of the two components found via Fast Independent Component Analysis applied to the DAS between patients and control subjects. The same two clusters (red, and blue) were consistently found with a K-means score of -1.2. **E**) TOP, Alteration in dynamics between different groups of tumors. BOTTOM, Logistic regression between DAS and periventricular (PV) tumor patients. Qualitative inspection reveals a tendency for periventricular tumors to display slower BOLD dynamics, although they were found to be non-significant presumably due to the small number of samples (N=5, p=0.31, one-sided U-test).

References

- [1] Aerts, *et al.* Modeling brain dynamics after tumor resection using The Virtual Brain. *NeuroImage* **213**, 116738 (2020). <https://doi.org/10.1016/j.neuroimage.2020.116738>
- [2] Aerts, *et al.* Modeling brain dynamics in brain tumor patients using the virtual brain. *eNeuro* 5 (3) 0083-18 (2018) <https://doi.org/10.1523/ENEURO.0083-18.2018>
- [3] Aerts, *et al.* Pre- and post-surgery brain tumor multimodal magnetic resonance imaging data optimized for large scale computational modelling. *Sci Data* **9**, 676 (2022). <https://doi.org/10.1038/s41597-022-01806-4>
- [4] Stoecklein, *et al.* Resting-state fMRI detects alterations in whole brain connectivity related to tumor biology in glioma patients. *Neuro-oncology*, vol. 22, no. 9, p. 1388–1398, (2020). <https://doi.org/10.1093/neuonc/noaa044>
- [5] Nanning, *et al.* Distributed changes of the functional connectome in patients with glioblastoma. *Scientific reports*, vol. 10, no. 1, p. 18312, (2020). <https://doi.org/10.1038/s41598-020-74726-1>
- [6] Silvestri, *et al.* Widespread cortical functional disconnection in gliomas: an individual network mapping approach. *Brain communications*, vol. 4, no. 2, p. fcac082, (2022). <https://doi.org/10.1093/braincomms/fcac082>
- [7] Sullivan, *et al.* Directionally encoded color track density imaging in brain tumor patients: A potential applications to neuro-oncology surgical planning. *NeuroImage: Clinical*, vol. 38, 103412 (2023). <https://doi.org/10.1016/j.nicl.2023.103412>
- [8] Menze, *et al.* The multimodal brain tumor image segmentation benchmark (BRATS). *IEEE transactions on medical imaging*, 34(10), 1993-2024, (2914).
- [9] Calabrese, *et al.* The University of California San Francisco Preoperative Diffusive Glioma MRI. *Radiology: Artificial Intelligence*, <https://doi.org/10.1148/ryai.220058>
- [10] Ellingson, *et al.* Consensus recommendations for a standardized Brain Tumor Imaging Protocol in clinical trials. *Neuro-Oncology*, 17(9), 1188–1198, 2015 [doi:10.1093/neuonc/nov095](https://doi.org/10.1093/neuonc/nov095)
- [11] Weller, *et al.* European Association for Neuro-Oncology (EANO) guideline on the diagnosis and treatment of adult astrocytic and oligodendroglial gliomas. *The Lancet Oncology*, S1470-2045(17)30194-8 (2017)

[12] Weller, *et al.* EANO guidelines on the diagnosis and treatment of diffuse gliomas of adulthood. *Nature Reviews Clinical Oncology*, (2021) <https://doi.org/10.1038/s41568-023-00576-4>

[13] Schwenck, *et al.* Advances in PET imaging of cancer. *Nat Rev Cancer* **23**, 474–490 (2023), <https://doi.org/10.1038/s41568-023-00576-4>

[14] Constable, R.T., Spencer, D.D. Repetition time in echo planar functional MRI. *Magnetic Resonance in Medicine*. 46:748-755 (2001) <https://doi.org/10.1002/mrm.1253>

[15] Murphy, *et al.* How long to scan? The relationship between fMRI temporal signal-to-noise ratio and necessary scan duration. *NeuroImage*, 34(2):565-574 (2007).

[16] Welvaert M., Rosseel Y. On the Definition of Signal-To-Noise Ratio and Contrast-To-Noise Ratio for fMRI Data. *PLoS ONE* 8(11): e77089. (2013) [doi:10.1371/journal.pone.0077089](https://doi.org/10.1371/journal.pone.0077089)

Reviewer #2

The authors present a carefully conducted study of tumor impact on the brain in relation to surgical intervention using structural and functional MRI data and machine learning methods. The authors demonstrate another approach to measure the impact of tumor and neurosurgery on brain function and structure.

The manuscript is carefully written. However, due to the use numerous methods, it is very long and addresses many aspects which makes it difficult to read.

Major

1. The authors claim that a new method for fiber tracking pipeline can use anatomical information and still reconstruct fiber bundles in tumoral and peritumoral tissue. However, this is only weakly clarified and illustrated in the manuscript.

(A & C) We appreciate the reviewer's general comment on the way the workflow is presented in the manuscript. As a general reply, we rewrote the subsection in the Methods of *Reconstruction of tumoral structural brain networks* in its entirety. In relation to this change, we have modified Figs. 3 and S8 and added 4 new examples in Figs. S11-S14. All the corresponding captions have also been updated and can be found at the end of this rebuttal. The subsection *Structural brain networks containing intra-tumoral fibers* has also been partially rewritten to adjust to the changes mentioned above as well as to reply to several concerns raised in this rebuttal. The subsection *Tracking fibers in the presence of brain tumors* has been adequately updated.

The changes incorporated hopefully strengthen the manuscript's readability and transparency. In the subsequent replies, all the implemented modifications are detailed.

2. The authors included a low number of tumor patients. The results are interesting regarding tumor patients. However, how do the authors justify the study design of a mixture with a completely different tumor entity, meningiomas, which do not grow from glial cells, and in which the influence of the tumors on white matter is different? Furthermore, gliomas 1-4 also differ significantly, be it in terms of kinds, aggressiveness, or growth potential.

(A) We agree with the concern raised by the reviewer that we considered during the early stages of the research. The dataset we studied was the only one with functional and diffusion MRI acquisitions that were publicly available. We failed to find more available data with the same or at least similar characteristics. Low sample sizes are common in studies on brain tumors [1-7] in reputable journals, and heterogeneity is ubiquitous [7,8] even lacking baselines, further compromising statistical inferences – usually corresponding to a healthy cohort.

Importantly, HARDI multi-shell diffusion MRI and functional MRI are not standardized protocols according to recent guidelines from the National Cancer Institute [9] and the European Association for Neuro-Oncology [10,11]. These factors pose a great challenge in the creation of large datasets, which would, of course, be desirable.

We note that we have conducted analyses for several groups to try to overcome the heterogeneity (see for example, Fig. 2 and Fig. 5). We are aware that grading and histology have different growing mechanisms, evolution, and prognostics. However, we took special care in not stating any clinical guidelines to follow but rather tried to provide a comprehensive study of the relationship between MR signals within the lesions and the brain at rest.

(C) We have explicitly added other studies with low sample size in *Limitations and Future Directions* as references, as well as the following line regarding the heterogeneity of the sample: “[...] Heterogeneity could also impact the results. However, to overcome the heterogeneity present in the data, we conducted some independent analyses for several groups. [...]”.

3. The authors claim that “The resulting brain networks captured streamlines and connections which were previously non-existent when using each algorithm alone (Fig. S6A).”

(A) We appreciate the reviewer’s thorough concern since this is one of the contributions we wanted to highlight. As stated before in our answer to the first comment, however, we do not wish to claim or produce an overstatement. Since there are several issues of each concern, we answer them separately below.

Importantly, after changing the streamline filtration method following the reviewer’s suggestion (i.e., comment 3 in the Minor section), we opted to slightly modify the end step of the pipeline and merge the connectivity matrices instead of the tractograms (see our reply to point 3D).

A new table with all the relevant parameters and their values has been added to the *Methods* section. Table 2 and its caption are attached at the end of this rebuttal for transparency.

A) I strongly doubt that msmt won’t reveal FOD’s within the tumor region. In Fig. 3, the authors show a brain mask which excludes the tumor (regarding their use of msmt). If they passed a 5TT image without a lesion mask, then they would obtain also FOD’s in the voxels of the tumor, cf. my other comment in the Minor section.

(C) We illustrate this point better in our reply to comments 5 and 11 in the Minor section. Briefly, we cannot confirm that this reviewer’s point on msmt holds true for each situation. As suggested, we ran the traditional multi-shell multi-tissue reconstruction algorithm **without**

masking the lesion and observed that the opposite holds true. In fact, Figs. S8, S11, and S12 show how the lesion greatly hampers the performance of the MSMT reconstruction algorithm. The newly updated Fig. 3B shows the same problem but from a “streamline” point of view. The fibers are stopped in the surrounding areas of the lesion due to a lack of anisotropic diffusion signal.

(C) We included the figures mentioned above in the manuscript and the supplementary material: Fig. 3, S8, Fig. S11, and Fig. S12 are new. The caption of Fig. S8 specifically addresses this problem: “Even if the tumor is not masked, the algorithm damps the anisotropic component of the diffusion signal.”.

Furthermore, we explicitly mention this issue in the results and discussion sections:

“As suspected (Aerts, et al. 2019), the multi-shell approach caused overdamping of the diffusion signal within the lesion mask (Fig. S8). In combination with this, the altered anatomical properties surrounding the lesion imposed an artificial early stop to streamlines that could, if diffusion signal was available, penetrate or circumvallate the tumor (Fig. S8).”

“Moreover, in the presence of pathological tissue, state of the art fiber reconstruction methods suffer from cerebrospinal fluid contamination due to the inflammation of neuronal tissue. Novel algorithms based on high angular resolution imaging are specifically designed to minimize the effects of this contamination, but neglect substantial information (i.e., multi-shell diffusion data) (Aerts, et al. 2019).”.

B) Why do the authors merge to different algorithms to estimate response functions? I suggest using one, e.g. MSMT_5TT or dhollander or then SS3T to have the same underlying response function in the resulting tractogram.

(A) We would like to clarify that we are using a unique response function estimated with data from the tissue outside the lesion (i.e., healthy tissue). We use the dhollander algorithm. The fiber orientation distribution functions (FODfs) are then reconstructed with the same underlying response function for both the tumor region (with SS3T) and the healthy tissue (with MSMT).

(C) The above has been made explicit in the *Methods* section of the manuscript as follows: “[...] Within this mask, for each b-value shell and tissue type (white matter, gray matter, and cerebrospinal fluid) a response function was estimated (Dhollander, et al. 2016); and the fiber orientation distribution functions (FODs) were built and intensity normalized using a multi-shell multi-tissue (MSMT) constrained spherical deconvolution approach (Jeurissen, et al. 2014). [...]”; and w.r.t. the lesion, “[...] These FODs were reconstructed, and intensity normalized only inside the lesion mask using the same underlying response function as estimated earlier in the healthy tissue. [...]”.

C) Please also explain the rationale for the usage of SS3T via one extracted shell of the multi-shell data since it is designed for single shell data, while all the shells would be available.

(A) Inside the tumor, the multi shell algorithm DOES not allow for an adequate estimation of the FODs (*Linked to comment 3A Major and 5 Minor*). A possible solution is free water fraction estimation (isotropic, similar to GM). Another one is to use the ss3t discarding some shells to obtain an underlying anisotropic FOD for the WM signal. It is only used INSIDE the lesion, hence not completely discarding the goodness of multi-shell acquisitions. To our

knowledge, this is the first attempt that tries to maintain as much data as possible (i.e., HARDI plus multi-shell) while still paying attention to the signal in the tumor.

(C) We included a statement about this point in the discussion: “In this regard, we designed a hybrid pipeline to use each method where it is best suited (Fig. S8) while, if available, using all the diffusion shells acquired. The SS3T method needs to be used only inside the tumor hence only discarding multiple diffusion shells in the lesion area. Further research should try to assess whether independently running SS3T with each shell can be combined or not.”.

D) Performing tractography exclusively within the tumor region is controversial. How are these streamlines controlled? I propose to perform tractography with FOD datasets that are not constrained by a tumor mask or combined and to carefully examine the FOD's in the tumoral region so that signal estimation considers the whole brain and only one response function is used.

(A & C) We agree with the reviewer that the steps in the previous version of the manuscript did not address the issue pointed out here thoroughly enough; therefore, we have addressed this issue by updating a few steps of the process. After changing the streamline filtration method to SIFT2, streamlines are now controlled in two different ways. The details of each way follow:

1) The number of streamlines generated within the lesion corresponds with the expected number of streamlines present in healthy subjects. More specifically, the average number of streamlines intersecting the masked lesion in the healthy cohort is used as a proxy for the patient's lesion tractograms. In simplified mathematical terms:

$$N_{streamlines\ in\ lesion} = \frac{1}{N_{control}} \sum_{i=1}^{N_{control}} \sum_{streamline=1}^{streamlines} 1 \cdot \begin{cases} 1 & \text{if } streamline \in \text{Lesion} \\ 0 & \text{otherwise} \end{cases}$$

This is stated in the *Methods* section of the manuscript as: “The number of streamlines was set as the average number of streamlines intersecting the lesion in the healthy cohort. We superimposed the lesion on the tractograms of each control subject and tallied the overlapping streamlines (Tournier, et al. 2019). This was important given that each lesion was in a different location and the natural density of streamlines in that specific location differed. This subject-specific streamline count controlled that the tract densities were comparable to the healthy cases (Fig. 3B-D; see also Fig. S11-S14).”.

2) The usage of SIFT2 allowed for a straightforward weighting of each streamline lesion and its final contribution to the connectome. Importantly, even if a single streamline has a trajectory extending beyond the tumor, only its contribution to the diffusion signal inside the lesion should be considered. Importantly, the regularized weight is constant throughout the trajectory irrespective of the mask where SIFT2 acts. That is, we applied SIFT2 using only the normalized white-matter FODs inside the lesion. This is stated in the section *Reconstruction of tumoral structural brain networks* as follows: “Spherical-deconvolution informed filtering (Smith, et al. 2015) was applied to ensure that each streamline adequately contributed to the *lesioned* diffusion signal (i.e., filtering was applied inside the lesion mask).”.

Moreover, our answer to comment 7A in the Minor section is also pertinent here. In that, we argue how the generated streamlines within the lesion are necessary to avoid early stop of longer fibers passing through the surrounding areas of the tumor (see also Fig. 3C). As also

discussed in comment 7B in the Minor section, these two steps effectively controlled the densities of streamlines closely matching that of healthy individuals and tissues.

As a last remark, we did not perform a tractography exclusively within the tumor region. We would like to insist that the usage of the multi-shell multi-tissue algorithm alone is not sufficient to deal with tumoral and peritumoral FODs and streamlines. This point is, we hope, thoroughly illustrated throughout this rebuttal: points 3A Major, 5 Minor, and 11 Minor as well as the updated figures 3, S8, S11-14. Moreover, as written in point B, we are using a **single response function** for the reconstruction of all types of FODs – both SS3T and MSMT use the same underlying response function. Lastly, the more intricate nature of the lesion naturally caused the FOD amplitudes to be smaller (Fig. S8, S11, and S12) – this was also observed in previous work [3].

4. The authors use two different atlases to obtain functional (Gordon parcellation) and structural (AAL) connectomes. Have they considered to use another atlas that would fit both acquisition schemes?

(A) We acknowledge the reviewer's comment even though we did not see a clear concern about the parcellations we used. Both the Gordon and the AAL are commonly used parcellations. Brain networks derived from the AAL parcellation only make sense when considering structural connections. In contrast, brain networks derived from the Gordon parcellation should only be considered when studying functional connectivity. The Glasser parcellation [12], for example, combines both modalities and, in theory, allows its use for multimodal studies. However, we point out that we are not considering information from function and structure jointly that would have required a different parcellation scheme as suggested by this reviewer.

While it is true that the structural prediction and study we conducted in this article could have used the Glasser parcellation, this would have introduced yet more issues. The Glasser parcellation considers twice as many regions as the AAL without incorporating subcortical regions. It was desirable to consider subcortical white matter pathways given that tumors and their effects need not be restricted to cortico-cortical connectivity.

Increasing the number of ROIs would have created more problematic fits of the machine learning models (e.g., larger number of model parameters), especially for the relatively low sample size. Furthermore, in the generative AI literature, the size of the brain networks is considerably lower than the ones we chose (e.g., all the references in the main text – in the introduction).

(C) We hope that we have addressed this reviewer's point and thus have made any changes to the manuscript related to this issue.

5. I miss in the abstract a short info about the result that the frequency domain showed local and distributed anomalies in time series.

(A) We thank the reviewer for this suggestion. We have expanded on this point in the *Abstract* and in the initial paragraph of the *Discussion*.

(C) In the abstract, we have rephrased the sentence present in line 19 of the unrevised version to the following: “[...] First, we find **intertwined alterations in the frequency domain of**

local and spatially distributed resting-state functional signals, potentially arising within the tumor. [...].”

Moreover, we have added the following sentence at the beginning of the *Discussion* section: “[...] Importantly, unlike previous studies, we focus on the signal present within the lesion and how it explains connectivity reorganization in both modalities. [...]”.

Minor

1. The authors do not provide the degrees of freedom regarding the correlation test results. Please provide all information.

(A) We thank the reviewer for this appreciation. For the functional part of the study N=25 thus 23 degrees of freedom, while for the structural N=19 (pre- and post-surgery pairs) thus 17 degrees of freedom.

(C) In all the places where we report correlation results, we included the degrees of freedom together with the correlation and significance values: “[...] (r=XX, df=YY, p=ZZ, test used)”.

2. Unfortunately, the authors also only provide p-values for their ANOVA and do not provide the entire picture. The authors need to provide the degrees of freedom (between groups, within groups) and the F value, cf. also t-tests.

(A) We appreciate this detail. We now include degrees of freedom and F values in the main text.

(C) This small change affects the section “*Structural predictions after tumor resection*” in the Results section.

3. There is a newer, improved version, for tractogram filtering, SIFT 2, as the authors are certainly aware of. Why did the authors use the old SIFT version?

(A) As pointed out correctly, we are aware of the existence of SIFT2. However, there is evidence that SIFT also enables accurate reconstruction of fiber tracts, with positive effects derived from its use [13]. Even in the original proposal, the authors acknowledged that SIFT is a valid method to use (see Discussion) and that SIFT2 addresses, above all, the high computational demands of the original proposal [14]. SIFT required the generation of extremely large numbers of streamlines to reach reliable tractograms after the filtration process. We noted that the heuristics behind SIFT are a bit more straightforward given that streamlines are removed instead of weighted to achieve greater correspondence between the tractogram and the underlying diffusion signal. Furthermore, very recent works in the field of diffusion/tractography have successfully used the first version of this filtering method (see for example, [15,16]).

(C) Nonetheless, we have changed all references to the SIFT method to the upgraded version SIFT2: In the caption of figure 3 and in the Methods (section of *Reconstruction of tumoral structural brain networks*). The regularization parameters have also been included in the text and the rationale for their choice (i.e., based on the SIFT2 L-curve optimization recommendations [14]).

This change led us to rethink the merging strategy since the tractograms remained unfiltered. Hence, instead of merging the streamlines into a single tractogram, we filtered the two reconstructed streamlines separately and merged the connectivity matrices (*See our*

reply to comment 3 Major).

4. What is the rationale for selecting 4M streamlines with an atlas consisting of 166 nodes? Have the authors tested reproducibility with sifted 4M streamlines and 166 nodes? I suspect that the number of streamlines needs to be increased to get reproducible tractograms.

(A) We agree that the reproducibility of the tractograms should always be taken into consideration. Previous studies have been conducted with 1M streamlines remaining after filtration [15]. It is true that with the original use of SIFT, we opted for a rather aggressive filtering threshold (~30%) when most of the literature has settled around lower more conservative values (e.g., ~10%) at the cost of increased computational time. Yet, a total number 1M streamlines after filtration and a reasonable maximum spherical harmonic coefficient (close to the default values, $L=8$) is considered as a safe practice to ensure high reproducibility [16].

However, we increased the number of streamlines to 8M trying to achieve a compromise between the computational feasibility and reproducibility of the obtained networks. We note that, following comments #3 Major and Minor from the reviewer, we did not filter the tractograms using SIFT thus not obtaining a “second” tractogram with a reduced number of streamlines.

Following the guidelines from Roine, *et al.*, we think that there is no clear concern about the size of the network we used (i.e., number of ROIs). Reproducibility with the stream counts we used, together with the changes we have incorporated after the reviewer’s suggestions, has also been studied for robustness for parcellations of similar characteristics [16].

(C) We have updated the Methods (section: Reconstruction of tumoral structural brain networks) to correctly include the final number of streamlines (i.e., 8M).

5. The authors combine the FOD’s obtained from MSMT with a tumor mask and SS3T within the tumor mask. How different is the MSMT signal from MSMT without a tumor mask? MSMT is stable, especially with the multi-shell dMRI acquisition and will also provide FOD’s in the peritumoral area.

(A) We agree with the reviewer’s inquiry. In fact, this very same question is what drove the idea to design the presented workflow. We kindly refer the reviewer to earlier research on the same dataset, in which the authors in that work proved that MSMT alone was not able to resolve the anisotropic component of the diffusion signal [3].

However, in what follows, we provide a detailed description of the impossibility of MSMT to reconstruct any anisotropic component of the diffusion MR signal. We have included screenshots of an example patient (i.e., sub-PAT16). We hope to convince the reviewer that special care is indeed needed to bypass the problems outlined in the main text.

In A we show how the segmentation of different tissues automatically delineates a grey matter (Left) and white matter (Centre) around the tumor (Right). This poses a problem for streamlines to traverse the tumoral region. The second row (B) shows the MSMT signal reconstructed if masking the tumor (Left) and without masking the tumor (Centre). Even if the tumor is not masked, the multishell algorithm damps the anisotropic component of the diffusion signal. In C, we show the SS3T signal inside the tumor, which loosely corresponds to the oedemic diffusion signal. This proves that, even if we were only masking the necrotic area (dark grey tissue), the MSMT alone would not be able to reconstruct the diffusion signal within the lesion (*See also our reply to comment 11 Minor*).

It is also noticeable that the anisotropic component of the signal within the tumor is weaker than that of the one surrounding the lesion. The FODfs are scaled by a factor of 5 in all images, a detail of 3, and a maximum order for SH coefficients of 8 (in the MRtrix3 viewer).

(C) The above figure has been incorporated into Fig. S8. The caption has been properly updated, and it is left at the end of this rebuttal. We have added four more plots in the supplements with the corresponding captions (Fig. S11 to Fig. S14; see the end of this rebuttal).

6. There is a typo on Line 283 “.” .

(A) Thank you for noticing the mistake. We have corrected it.

7. Figure S6,

(A) We agree with the reviewer’s observation. The updates following the reviewer’s 3rd comment (Major and Minor) have yielded further insight into the density of streamlines within the lesions. Figure S6 in the unrevised manuscript has been properly updated and several new images are shown throughout the main text and the supplements. The new figures and captions are the following: Fig. 3B-C, S8, and S11-S14. We hope that these new points of view and further examples on different subjects illustrate the need for accurate fiber tracking in focal lesions. We reply to each comment present in this point separately.

A) How can the authors prove that these streamlines are not erroneous?

(C) This question is difficult to answer with quantitative arguments. We refer the reviewer to our answer in point 3D Major. We provide visual proof that the streamlines that are generated due to our method are natural continuations of the streamlines that are “cut” by the lesion. More specifically, in figures

- 3B and Fig. 3C, a large bundle of cortico-spinal tracts is shown that without special tracking in the peritumoral region is truncated potentially inflating the connectivity weights between regions of interest being located near the tumor. The caption of these figures state the following: “Importantly, large cortico-spinal and superior longitudinal fasciculus peritumoral fiber bundles are now visible”.

- 3D, the whole set of streamlines intersecting the tumors that are generated with our hybrid approach appears to be in correspondence with the reduced set that is found when only using a multi-shell multi-tissue approach.

- S13 and S14 the same phenomenon is observed. The lesion causes early stopping of fibers around the tumor and the anatomical constraints prevent these peritumoral fibers from penetrating the lesion.

We also quantify the differences in scale-free properties of the networks derived with our pipeline and a more classical multi-shell multi-tissue approach (without masking the tumor and only using MSMT). These analyses are now described in the Supplementary Material in the section *Weight probability distribution of brain networks*: “We adjusted a power law distribution to the weighted degrees (Alstott, et al. 2014),

$$P(x) \propto x^{-\alpha}, \quad (\text{S8})$$

where α is the power law exponent measuring the decay in the probability of finding strongly connected nodes in the network (e.g., hubs). We independently fit a power law distribution to each network derived from the healthy cohort and patient. Importantly, for each patient, we fit a power law distribution for the network obtained with our pipeline and another fit for the network obtained with a canonical multi-shell multi-tissue approach without masking the tumor. Next, we compared the differences between the exponent α for each pipeline and the healthy cohort and the differences between pipelines (see Fig. S8D). Importantly, we consider two different scaling ranges: $x_{min} = 100$ and $x_{min} = 300$, the latter one evaluating the scale-freeness in the large asymptotic range.”.

The results of these analyses have been added in the Results subsection *Structural brain networks containing intra-tumoral and peritumoral fibers*. This section has been rewritten in the revised version of the manuscript. Briefly, we state that streamlines generated follow natural white-matter tracts and that scale-free properties of the resulting networks are comparable between methods and differences between pipelines disappear for small tumors, thus suggesting that pipelines asymptotically converge for small tumors (as expected given that for small tumors, the lesion mask approximates an empty region). Also, see Fig. S8 and its caption (both copied at the end of this rebuttal document).

The following paragraph has been added to the discussion further emphasizing the “natural” extension of the peritumoral fibers that our approach generated, and a specific comment regarding this early stop of peritumoral fibers has been added in the results and discussion section given its potential implications for surgical planning and tumor resection: “In this direction, our pipeline used anatomical constraints outside tumors while entirely relying on

the diffusion signal inside the tumor. Importantly, the relaxation of the anatomical constraints in peritumoral tissue together with the incorporation of intra-tumor FODs, yielded well-known tracks that were otherwise truncated (Fig. 3B-C; see also Fig. S13-S14). Brain tumors are heterogeneous in size and location; thus, it is important to emphasize that our pipeline gradually reduces to a “standard” setting the smaller the tumor is. We studied the scale-free properties of the structural networks derived from our pipeline and observed that they asymptotically converged to the properties of networks derived without considering peritumoral fibers (Fig. S8D; see also the Supplementary Material). For example, if a tumor were of small size and located mostly within gray matter, most of the peritumoral fibers would remain unaffected. However, in more critical cases, such as the ones shown here, the artificial truncation of streamlines derived from canonical approaches results in a larger percentage of disconnections than our hybrid pipeline (Fig. 3D). Crucially, network neuroscience studies might rely on these artificial disconnections to report structural disconnections without truly assessing whether they are indeed present or introduced (Fekonja, et al. 2022). Furthermore, the deletion of important fiber tracks could have detrimental effects on surgical planning given the evermore common use of tractography in pre-operative settings (Sullivan, et al. 2023), needing personalized models that consider microstructural properties of the tumors for more accurate tracking alternatives (Nilsson, et al. 2018).”.

B) In Figure S6 part A the streamlines in the tumoral area appear much denser, as if there were much more streamlines as in the other WM areas, please explain.

(C) To avoid generating higher densities of fibers inside the lesion w.r.t. the healthy counterpart, we control the number of streamlines generated within the lesion and perform spherical-deconvolution informed filtering only using the FODs present inside the lesion. This ensures that the method does not oversample streamlines and sets an exact subject-specific number of streamlines for each lesion. In the main manuscript (see *Methods*) we explicitly describe this: “The number of streamlines was set as the average number of streamlines intersecting the lesion in the healthy cohort. We superimposed the lesion onto the tractograms of each control subject and tallied the overlapping streamlines (Tournier, et al. 2019). This was important given that each lesion was in a different location and the natural density of streamlines in that specific location differed. This subject-specific streamline count controlled that the tract densities were comparable to the healthy cases (Fig. 3B-D; see also Fig. S11-S14). Spherical-deconvolution informed filtering (Smith, et al. 2015) was applied to ensure that each streamline adequately contributed to the *lesioned* diffusion signal (i.e., filtering was applied inside the lesion mask).”. The new streamline densities closely match their healthy counterparts and even the densities obtained without our hybrid approach (see Fig. 3B-D, S13, and S14.

C) The yellow tumor mask obscures the streamlines. Please provide an image showing the coronal and sagittal aspects as well, preferably with a transparent tumor mask and in relation to the FOD's.

(C) All masks shown in the new figures are transparent and hopefully do not interfere with the streamlines. Furthermore, to provide a different perspective, in the supplementary figures we show directionality encoded track density maps while in the main text, we opted for the raw streamlines.

D) I couldn't find a description for Figure S6 part C – a modified 5TT image?

(C) We thank the reviewer for noticing. We have incorporated this step in the new Fig. 3A describing the proposed workflow and a sentence in the new *Methods* section: “The lesion mask was added to the 5 tissue-type image to be considered as pathological tissue (Tournier, et al. 2019)”.

8. MSMT and SS3T show different crops in Figure 3.

(A) We appreciate the suggestion. We have corrected this misalignment in the new figure and showed the same crops for both methods.

9. Which activation function was used for the hidden layer and the output layer? The model architecture is described sparsely.

(A) We thank the reviewer for noting the lack of clarity regarding the description of the model. We used a standard fully connected architecture comprising a single hidden layer with a sigmoid activation function.

(C) We have incorporated the above specifications in the section *Generation of post-surgery graphs with artificial neural networks* in the Supplementary text. The sentence reads as follows: “[...] of size $E/2$. Therefore, for each edge k in the n -th connectome, the likelihood in Eq. (S3) was obtained by propagating the input through the artificial network as follows:

$$\left\{ \begin{array}{l} r_{ni} = \sigma \left(\sum_{j=1}^E W_{ij}^{\text{input}} \cdot x_{nj} \right) \\ \mathcal{L}(y_{nk} = \epsilon \mid \mathbf{x}_n) = \sum_{i=1}^{E/2} W_{ki}^{\text{output}} \cdot r_{ni} \end{array} \right.$$

where $\sigma(x) = \frac{1}{1+e^{-x}} \in [0,1]$ is a sigmoid activation function. Note how the network compresses the information in the intermediate layer forcing a low dimensional representation of connectomes' features. The model weights W_{ij}^{input} and W_{ij}^{output} were updated using stochastic gradient descent to minimize the mean squared error between the connectomes generated by the model and the ones obtained by applying the tractography pipeline described in the main text. **Several** architectures [...]”.

10. Please clarify the pathologies of the patients used in Fig. 4.

(A) We thank the reviewer for noticing this lack of information.

(C) The pathologies of each patient shown in Fig. 4 have been incorporated in the corresponding caption: “[...] PAT03: meningioma grade I, location parietal right, 78.44 cm³. PAT15: meningioma grade I, location frontal right, 2.13 cm³ PAT28: oligodendroglioma grade II, location frontal left, 11.49 cm³.”.

11. I would prefer to see more visual results regarding the obtained FOD's in the tumoral areas and a comparison to the use of MSMT response function in the tumoral area to visually inspect differences to the results from SS3T. Without this, the complicated additional effort with the combination of another response function is presented to the readership in a way that is insufficiently transparent.

(A) This same phenomenon was thoroughly investigated in an earlier study [3]. Our contribution lies in the usage of multishell data even in this scenario where single shell data is required to “dig” inside the lesion (*Related to points 3A and C Major as well as 5 Minor*).

We would like to emphasize that even if the MSMT was able to recover the diffusion signal inside the lesion (at least partially), the anatomical constraints would stop the fibers around the peritumoral area (see new Fig. 3 and Fig. S8), hence further justifying an alternative approach to tractography in the presence of lesions. Whether the same scenario applies to other focal lesions or not (i.e., stroke), remains an open question. In what follows, we illustrate this by obtaining directionally encoded color track density maps of the streamlines that intersect the lesion with our two methods. It is visible that, if solely using MSMT, fibers would not be able to penetrate the lesion, potentially losing valuable information for surgical planning.

(C) These examples, in addition to the figure used to reply to points 3 (Major), and 5 (Minor), have been added in the supplements with the corresponding captions (i.e. Fig. S11 to Fig. S14). These figures are also attached at the end of this rebuttal.

References

- [1] Aerts, *et al.* Modeling brain dynamics after tumor resection using The Virtual Brain. *NeuroImage* **213**, 116738 (2020). <https://doi.org/10.1016/j.neuroimage.2020.116738>
- [2] Aerts, *et al.* Modeling brain dynamics in brain tumor patients using the virtual brain. *eNeuro* **5** (3) 0083-18 (2018) <https://doi.org/10.1523/ENEURO.0083-18.2018>
- [3] Aerts, *et al.* Pre- and post-surgery brain tumor multimodal magnetic resonance imaging data optimized for large scale computational modelling. *Sci Data* **9**, 676 (2022). <https://doi.org/10.1038/s41597-022-01806-4>
- [4] Stoecklein, *et al.* Resting-state fMRI detects alterations in whole brain connectivity related to tumor biology in glioma patients. *Neuro-oncology*, vol. 22, no. 9, p. 1388–1398, (2020). <https://doi.org/10.1093/neuonc/noaa044>
- [5] Nenning, *et al.* Distributed changes of the functional connectome in patients with glioblastoma. *Scientific reports*, vol. 10, no. 1, p. 18312, (2020). <https://doi.org/10.1038/s41598-020-74726-1>
- [6] Silvestri, *et al.* Widespread cortical functional disconnection in gliomas: an individual network mapping approach. *Brain communications*, vol. 4, no. 2, p. fcac082, (2022). <https://doi.org/10.1093/braincomms/fcac082>
- [7] Sullivan, *et al.* Directionally encoded color track density imaging in brain tumor patients: A potential applications to neuro-oncology surgical planning. *NeuroImage: Clinical*, vol. 38, 103412 (2023). <https://doi.org/10.1016/j.nicl.2023.103412>
- [8] Pernet, *et al.* A structural and functional magnetic resonance imaging dataset of brain tumour patients. *Sci Data* **3**, 160003 (2016). <https://doi.org/10.1038/sdata.2016.3>

[9] Ellingson, *et al.* Consensus recommendations for a standardized Brain Tumor Imaging Protocol in clinical trials. *Neuro-Oncology*, 17(9), 1188–1198, 2015
[doi:10.1093/neuonc/nov095](https://doi.org/10.1093/neuonc/nov095)

[10] Weller, *et al.* European Association for Neuro-Oncology (EANO) guideline on the diagnosis and treatment of adult astrocytic and oligodendroglial gliomas. *The Lancet Oncology*, S1470-2045(17)30194-8 (2017)

[11] Weller, *et al.* EANO guidelines on the diagnosis and treatment of diffuse gliomas of adulthood. *Nature Reviews Clinical Oncology*, (2021) <https://doi.org/10.1038/s41568-023-00576-4>

[12] Glasser, *et al.* A multi-modal parcellation of human cerebral cortex. *Nature* **536**, 171-178 (2016) <https://doi.org/10.1038/nature18933>

[13] Smith, *et al.* The effects of SIFT on the reproducibility and biological accuracy of the structural connectome. *NeuroImage* **104**, 253-265 (2015)
<https://doi.org/10.1016/j.neuroimage.2014.10.004>

[14] Smith, *et al.* SIFT2: Enabling dense quantitative assessment of brain white matter connectivity using streamlines tractography. *NeuroImage* **119**, 338-351 (2015)
<https://doi.org/10.1016/j.neuroimage.2015.06.092>

[15] Basile, *et al.* White matter substrates of functional connectivity dynamics in the human brain. *NeuroImage* **258**, 119391 (2022) <https://doi.org/10.1016/j.neuroimage.2022.119391>

[16] Roine, *et al.* Reproducibility and intercorrelation of graph theoretical measures in structural brain connectivity networks. *Medical Image Analysis* **52**, 56-67 (2019)
<https://doi.org/10.1016/j.media.2018.10.009>

Updated tables and figures

Tractography parameter	Value
Maximum SH coefficient order (l_{max})	8
Number of seeding points	80 – million
FOD amplitude cutoff	0.08
Min/max streamline length	3/280 – voxel size units
Euler integration step size	0.5 – voxel size units
Maximum angle	45 – degrees
SIFT2 Tikhonov regularization	0.1
SIFT2 Asymmetric Total variance regularization	0.08
Total number of streamlines	8 – million

Table 2 Tractography parameters. A detailed summary of the parameters and values used for all fiber orientation distribution functions, tractography, and regularization steps. Note how the total number of streamlines is constant (i.e., 8 million) for all subjects but each subject has a unique number of streamlines being generated within the lesioned tissue thus altering the number of streamlines outside the lesion.

Figure 3 Reconstruction of brain structural networks. **A)** Summary of the workflow designed to generate the tractograms in the presence of a brain tumor. Multi-shell multi-tissue (MSMT) reconstruction methods are run together with anatomically constrained tractography (ACT) to obtain the “healthy” fibers without including lesioned tissue. Fiber orientation functions inside the lesions are obtained by running the single-shell 3-tissue (SS3T) method only inside the lesioned regions (Tumor mask). Fiber orientation functions (FODs) from both methods are then merged. Connections originating, terminating or traversing oedemic tissue are tracked with the iFOD2 probabilistic algorithm by only seeding inside the lesion. A maximum angle and FOD amplitude cutoff stopping criteria are used. Both tractograms are used to obtain a weighted connectivity matrix with the outputs of streamline filtering (i.e., SIFT2). As a final step, the *lesion* and *healthy* structural matrices are merged using a customized formula (e.g., greedy). **B)** Axial, coronal, and sagittal planes (thickness of 1mm) of the tractogram from sub-PAT16 obtained with a simple multi-shell multi-tissue without masking the tumor (see Methods). 1mm thick cropping point is $(x,y,z)=(35,-17,-3)$ mm in MNI space. **C)** Tractogram

from sub-PAT16 (identical coordinates and thickness as in **B**) but having used the hybrid method outlined in **A**. Importantly, large cortico-spinal and superior longitudinal fasciculus peritumoral fiber bundles are now visible. **D**) Overview of the whole brain tractogram of the same subject with a simple MSMT approach (top) and our hybrid pipeline (bottom).

Figure S8 Peritumoral anatomy and FODs. **A)** Binary mask depicting the gray-matter (LEFT), white-matter (CENTER), and tumor (RIGHT) tissues of sub-PAT16. The tissue surrounding the lesion is automatically labeled as gray-matter thus imposing an anatomical barrier to regular fiber tracking algorithms. **B)** Tumoral and peritumoral FODs reconstructed with the MSMT algorithm when explicitly masking the tumor (LEFT) and without masking the tumor (RIGHT). Even if the tumor is not masked, the algorithm damps the anisotropic component of the diffusion signal. **C)** Tumoral and peritumoral FODs reconstructed with the SS3T algorithm applied only inside the lesion mask. Only this algorithm can recover the diffusion signal, which in this example, loosely maps the inflammation caused by the tumor. **D)** Differences in the scale-freeness, as measured by the power law exponent, between the networks obtained with our pipeline and a canonical multi-shell multi-tissue approach (blue scatter plot) and between each pipeline and the healthy cohort (black boxplot). Green triangles depict the means, black horizontal lines depict the medians, and the boxes show the 1st and 3rd quartiles. Significance tests in green show the p-values for the differences between means. x_{min} refers to the minimum scaling range taken to fit the power law distribution.

Figure S11 Tumor FODf reconstruction in sub-PAT05. Shown slices correspond to the intersection point $(x,y,z) = (-36,1,-2)$ mm, or voxel $(55,127,70)$ in MNI space. MRtrix3's viewer FODf display settings are detail 3, maximum order 8, and scale 5. **A** Using MSMT without masking the tumor. **B** Using SS3T only inside the lesion.

Figure S12 Tumoral FODf reconstruction in sub-PAT29. Shown slices correspond to the intersection point $(x,y,z) = (-15,43,-20)$ mm, or voxel $(76,169,93)$ in MNI space. MRtrix3's viewer FODf display settings are detail 3, maximum order 8, and scale 5. **A** Using MSMT without masking the tumor. **B** Using SS3T only inside the lesion.

Figure S13 Tumor streamlines in sub-PAT05. Direction encoded color (DEC) track density maps shown correspond to point $(x,y,z) = (-36,1,-2)$ mm, or voxel $(55,127,70)$ in MNI space. **A** Using MSMT without masking the tumor. **B** Using SS3T only inside the lesion. DEC maps with an intensity threshold of $[3,298.36]$ in **A** and $[3,418.27]$ in **B**. Color bar of $[10,180]$ in MRtrix3's viewer.

Figure S14 Tumoral streamlines in sub-PAT29. Direction encoded color (DEC) track density maps shown correspond to point $(x,y,z) = (-15,43,-20)$ mm, or voxel $(76,169,93)$ in MNI space. **A** Using MSMT without masking the tumor. **B** Using SS3T only inside the lesion. DEC maps with an intensity threshold of $[3,298.36]$ in **A** and $[3,418.27]$ in **B**. Color bar of $[10,180]$ in MRtrix3's viewer.

Reviewer #3

In this study the authors investigated the impact of brain tumors on functional and diffusion MRI. They found that intra-tumor fMRI signals were correlated with alterations in healthy tissue and resting-state networks. From a structural view, they designed a hybrid tractography pipeline capable of combining anatomical constraints and diffusion signals inside tumors, and further proposed a machine learning method to predict postsurgical structural rewiring. In general, the topic is interesting, and the findings have important clinical implications for tumor surgery.

(A) We thank the reviewer for the positive feedback. We also acknowledge the pitfalls mentioned below. We provide our answers (A) in red and changes (C) in blue, hoping they suffice to strengthen and improve the overall manuscript.

However, I still have some comments as follows:

1. The major concern I have with this paper is the comparison of BOLD signals. The experimental group used segmentation masks covering all parts of the tumor, from necrotic tissue to oedema, and their BOLD signals were compared with the signals in the same regions of the control subjects. However, a tumor may consist of different tissues, i.e., edema and necrosis, and therefore exhibits different functional signals. Similarly, a region with a mixture of neurons and tumor cells may show different functional characteristics. The authors should explain the rationale of directly comparing the signals in the whole tumors with that in the normal tissues.

(A) We appreciate the reviewer's comment. Importantly, the lesion masks did not include multi-tissue labeling hence our approach to include all the lesions. However, we agree that brain tumors display complex microstructural anatomies. Therefore, we contacted two clinicians with +30 and +10 years of experience (who have been included as co-authors) to perform and independently validate the segmentations of the tumor core. As a result of the lack of other modalities than T1, only the tumor core was reliably identifiable.

(C) We applied the same pipeline and methods using the masks containing the "tumor core" and the masks containing the "non-necrotic" tissue for glioma-like tumors. The results remained (qualitatively unaltered) and are added as two supplementary figures, Fig. S6 and Fig. S7. The implications of these results are now discussed in the discussion, at the end of the subsection *Alterations in local oscillations are related to distributed functional desynchronization in brain tumor patients*. The added lines are the following: "The relationship between the alterations in the DMN and the tumor remained unchanged when considering only the tumor core or the non-necrotic tissue of the lesion for patients with gliomas (Fig. S6-7)." The captions of these two new figures have also been added and can be found at the end of this rebuttal.

The methods section has been updated with the steps followed to obtain the segmentations for the tumor core and non-necrotic regions for glioma-like patients: "To identify the tumor core of gliomas, two clinicians with more than thirty and ten years of experience performed and independently validated the segmentations using 3D slicer. Data only allowed for the identification of the tumor cores; hence we subtracted the resulting cores from the whole lesion to obtain a "non-necrotic" region for each of the patients diagnosed with a glioma-like tumor."

Lastly, to be consistent and when appropriate, we have changed the words "oedema" and "oedemic" for "tumor" or "tumoral" in some of the subsections' titles. Most importantly, in the Methods subsection *Reconstruction of tumoral structural brain networks*, several of the aforementioned changes have been made.

2. There are many subjective perspectives in this paper that should be removed. For example, in line 110, "Fast and naïve inspection of those signals did not yield visual differences between tumoral and healthy signals"; in line 153, "visual differences between functional matrices (i.e., connectivity strengths) seemed to be anticorrelated with the magnitude of the DAS". The authors should focus on the findings supported by their data.

(A) We agree with the reviewer to remove subjective perspectives from the text.

(C) Therefore, the following changes have been made:

- “Fast and naïve inspection of those signals did not yield visual differences between tumoral and healthy signals (Fig. 1A LEFT).”, located at line 110 in the unrevised version, **has been deleted**. We have added the reference to Fig. 1A LEFT in the previous sentence, that is, “the same region in control subjects (Fig. 1A LEFT, see also Methods).”
- “However, visual differences between functional matrices (i.e., connectivity strengths) seemed to be anti-correlated with the magnitude of the DAS.”, located at line 153 in the unrevised version, **has been deleted**.
- Part of the sentence located at line 121 in the unrevised version has been deleted as follows. “The autocorrelation functions between time signals **were more easily comparable, since they** exhibited similar shapes for short time lags (Fig. 1B inset).”

3. In line 113, the authors should clarify whether the DMN signals are from the same patient or from a healthy control.

(A) We emphasize that these DMN signals are NOT from the same patient but rather from the control (healthy) cohort.

(C) To clarify this issue, we have rephrased the whole idea. The sentence “As such, we qualitatively compared Blood-Oxygen-Level-Dependent (BOLD) signals from damaged tissues with signals from regions belonging to the Default Mode Network (DMN) (Fig. 1A RIGHT, see also Methods).”, located at line 111 in the unrevised version, has been changed to “There were also no straightforward differences concerning Blood-Oxygen-Level-Dependent (BOLD) signals from regions belonging to the Default Mode Network (DMN) in the control group (Fig. 1A RIGHT, see also Methods).”.

4. In the first part of the results, please explain why the signals inside tumors are compared to the signals in the DMN but not in other regions?

(A) This is an important point in our analysis. We acknowledge that it was not made clear enough, thus we appreciate the reviewer for pointing it out. When addressing the question of whether a BOLD signal was present in the tumor area, we were faced with the question “what does it mean to be BOLD active”. We found BOLD oscillations inside the masked lesion (Fig. 1A) both in patients and in healthy subjects. That is, for each patient, the corresponding lesion was superimposed on every healthy subject for direct comparison.

One of our research questions was to study the relationship between BOLD activity within the lesion and in resting-state activity. Hence, we first benchmarked how functional signals behaved in regions that are known to be functionally relevant during subject resting, that is, the Default Mode Network [1].

(C) We have rephrased the beginning of the second paragraph of the results section in the hope that it addresses the concern more clearly. We have rewritten it as follows: “We were interested in characterizing the relationship between BOLD oscillations present inside the masked lesion and brain areas known to be functionally active and coherent (i.e., belonging to DMN regions) in resting conditions [32]. Thus, to test whether and how functional activity inside tumors was related to global signals, we first studied how brain tumors themselves shaped those global signals.”.

5. In Table 1, please explain what the bold numbers represent?

(A) We thank the reviewer for carefully noting this issue.

(C) We have added the following sentence in the caption of the mentioned table and all those found in the Supplements. “Bold numbers indicate better performance; lower is better for MSE, MAE, KL, and JS while higher is better for CS and PCC.”

6. In Fig. 5B left, the r values shown in the figure are inconsistent with the text, please explain.

(A) We are very grateful for the spotting of this inconsistency. During the writing of this manuscript, we performed several tests (as it is common practice) to study both the robustness of the effects as well as the validity of our claims. In one of these tests, we dropped different patients (the 4th in terms of size) and obtained the results that were written in the text. However, we found the reason to drop an additional patient to be unexplanatory. The figure shows the correct correlation coefficient for the colors depicted in it. We apologize for this lack of clarity.

(C) Therefore, we have corrected the correlation coefficients and their p-values accordingly. The older values were: ($r=-0.187$, $p=0.22$, one-tailed t-test); while the correct values are: ($r=-0.336$, $p=0.04$, one-tailed t-test).

7. In line 375, why tumors < 12.95cm³ are defined as small tumors, and tumors larger than that are large tumors? Is there any reference for this criterion?

(A) We would like to clarify this. We agree that group divisions should be justified either by clinical relevance or by data structure. The reference for this subdivision is the median of the dataset (the percentile 50) which, for this specific set, is equal to 12.95cm³. While we acknowledge that different criteria might be equally applicable, we found that this one was used in similar analyses [2] – although in different neuroscience fields.

(C) To further emphasize the selected criterion, we have specified the exact value of the median when defining the criterion. The sentences in the specified line now read: “[...] into two groups using the median of the whole dataset (Fig. 5B RIGHT; $P50=12.95\text{cm}^3$). Subjects with small tumors (size < median) showed [...]”. We have also added this number in the caption of Fig. 5B RIGHT and changed it as follows: “The subdivision was made based on the median of the set ($P50=12.95\text{cm}^3$).” instead of simply writing “[...] (Percentile 50)”.

References

[1] Buckner, R.L., DiNicola, L.M. The brain’s default network: updated anatomy, physiology and evolving insights. *Nat Rev Neurosci* **20**, 593–608 (2019). <https://doi.org/10.1038/s41583-019-0212-7>

[2] Sarno, *et al.* 2022. Dopamine firing plays a dual role in coding reward prediction errors and signaling motivation in a working memory task. *Proc. Nat. Acad. Sci.*
<https://doi.org/10.1073/pnas.21133111>

Updated figures

Figure S6 DMN and tumor core BOLD signals. **A)** LEFT Scatter plot showing the null correlation of DAS with distance and overlap between the tumor core and the DMN centroids. RIGHT, A strong linear trend was found between alterations inside the patients' tumor core and alterations in the DMN as measured by the DAS. The orange line corresponds to the significant linear fit (two-sided exact test), and shaded areas mark the 95% confidence interval. **B)** Differences in total power relative to healthy subjects (LEFT) and dynamics (RIGHT). Only the absolute values were significantly different from zero (one-side t-test and U-test). **C)** Scatter plot of the two components found via Fast Independent Component Analysis applied to the DAS between patients and control subjects. The same two clusters (red, and blue) were consistently found. **D)** TOP, Alteration in dynamics between different groups of tumors. BOTTOM, Logistic regression between DAS and periventricular (PV) tumor patients. Qualitative inspection reveals a tendency for periventricular tumors to display slower BOLD dynamics, although they were found to be non-significant presumably due to the small number of samples ($N=5$, $p=0.33$, one-sided U-test).

Figure S7 DMN and non-necrotic tumor BOLD signals. **A) LEFT**, Scatter plot showing the null correlation of DAS with distance and overlap between the non-necrotic tissue and DMN centroids. **RIGHT**, A strong linear trend was found between alterations inside the patients' non-necrotic tissue and alterations in the DMN's as measured by the DAS. The orange line corresponds to the significant linear fit (two-sided exact test) and shaded areas mark the 95% confidence interval. **B)** Differences in total power relative to healthy subjects (LEFT) and dynamics (RIGHT). Only the absolute values were significantly different from zero (one-side t-test and U-test). **C)** Scatter plot of the two components found via Fast Independent Component Analysis applied to the DAS between patients and control subjects. The same two clusters (red, and blue) were consistently found. **D) TOP** Alteration in dynamics between different groups of tumors. **BOTTOM** Logistic regression between DAS and periventricular (PV) tumor patients. Qualitative inspection reveals a tendency for periventricular tumors to display slower BOLD dynamics, although they were found to be non-significant presumably due to the small number of samples ($N=5$, $p=0.28$, one-sided U-test).

Reviewers' comments:

Reviewer #1 (Remarks to the Author):

The authors' responses are detailed and demonstrate a concerted effort to address each concern raised, ranging from clarifying the logical thread of the paper to addressing methodological issues such as sample size, data inconsistency, and statistical analysis choices. They have made changes to the manuscript to improve clarity, provided additional information where necessary, and justified their methodological decisions with references to existing literature and detailed explanations of their approaches. The authors have also acknowledged limitations and made efforts to mitigate potential biases or confounding factors, incorporating additional analyses and adjustments based on the feedback. Overall, the response is thorough, showing the authors' commitment to enhancing the quality and reliability of their work, some concerns may still linger.

3 In reviewing the manuscript, I noted the addition of information regarding the time intervals between scans, determined solely by clinical practice. However, the description does not explicitly address whether there are inter-group differences in these intervals for the three groups studied. If present, such differences could introduce a confounding variable, potentially impacting the study's findings by affecting the comparability of results across groups. For example, variations in recovery times, the progression of pathological changes, or the effects of treatment could differentially influence the results, thereby affecting the study's conclusions.

4 In response to the challenges posed by inconsistent fMRI parameters, an alternative strategy might involve excluding this data to re-analyze the remaining dataset. This approach, aimed at enhancing data integrity, warrants careful consideration of its potential downsides, such as the reduced sample size's impact on statistical power and the breadth of conclusions. I recommend a supplementary analysis, accompanied by a detailed exploration of these aspects, comparing outcomes from the full dataset against those from the refined subset. This dual analysis should illuminate the robustness of your findings, offering a clearer understanding of their reliability. Documenting this process in the supplementary materials will significantly bolster the credibility and depth of your research.

Reviewer #2 (Remarks to the Author):

The authors have addressed all my comments and improved the manuscript significantly.

Reviewer #3 (Remarks to the Author):

The authors are responsive and have addressed all my concerns.

Reviewer #1 (Remarks to the Author):

The authors' responses are detailed and demonstrate a concerted effort to address each concern raised, ranging from clarifying the logical thread of the paper to addressing methodological issues such as sample size, data inconsistency, and statistical analysis choices. They have made changes to the manuscript to improve clarity, provided additional information where necessary, and justified their methodological decisions with references to existing literature and detailed explanations of their approaches. The authors have also acknowledged limitations and made efforts to mitigate potential biases or confounding factors, incorporating additional analyses and adjustments based on the feedback. Overall, the response is thorough, showing the authors' commitment to enhancing the quality and reliability of their work, some concerns may still linger.

We appreciate the feedback about the revised version. Regarding the minor concerns remaining, we provide our answers (A) in red and changes (C) in blue in both this document and in the revised manuscript.

3. In reviewing the manuscript, I noted the addition of information regarding the time intervals between scans, determined solely by clinical practice. However, the description does not explicitly address whether there are inter-group differences in these intervals for the three groups studied. If present, such differences could introduce a confounding variable, potentially impacting the study's findings by affecting the comparability of results across groups. For example, variations in recovery times, the progression of pathological changes, or the effects of treatment could differentially influence the results, thereby affecting the study's conclusions.

(A) We thank the reviewer for this appreciation, and we agree that differences may have introduced effects. There are no inter-group differences regarding the intervals between the two MRI scans.

(C) We have added the appropriate information in the Methods section (Line 633): “There were no differences in the time intervals between the groups (meningioma [243±12 days], glioma [223±15 days], $p=0.328$, two-sided U-test).”.

We have also added a sentence acknowledging the potential factors that individual trajectory could introduce (Line 591): “Despite not showing inter-group differences in the time interval between scans, individual trajectories should also be considered, and thus investigated in more detail.”.

4. In response to the challenges posed by inconsistent fMRI parameters, an alternative strategy might involve excluding this data to re-analyze the remaining dataset. This approach, aimed at enhancing data integrity, warrants careful consideration of its potential downsides, such as the reduced sample size's impact on statistical power and the breadth of conclusions. I recommend a supplementary

analysis, accompanied by a detailed exploration of these aspects, comparing outcomes from the full dataset against those from the refined subset. This dual analysis should illuminate the robustness of your findings, offering a clearer understanding of their reliability. Documenting this process in the supplementary materials will significantly bolster the credibility and depth of your research.

(A) We thank the reviewer for this comment. We conducted the proposed analyses for two different results: the correlation between node similarity and alterations in the frequency spectrum and the relationship between local and global changes in the dynamics (i.e., Fig 2A LEFT and Fig. 2B RIGHT). These two analyses were the most critical to our interpretations. The results remained identical but only decreased in significance – as expected due to the reduced statistical power.

We have also slightly extended the proposal by conducting permutation analyses on different subsets of the data. Finally, we fitted an ordinary least squares model to the data explicitly accounting for the TR effects and we found them to be non-significant, as opposed to the significant effects of the DAS.

(C) All of the above has been added, as suggested, to the supplementary material in the form of a new subsection titled “Quantifying the effects of the Repetition Time (TR) on the reorganization of the functional networks”. This has been indicated in the main text as well (Line762). Additionally, a new figure (S15) and the corresponding caption have also been added and can be seen in this rebuttal.

Figure S15 Contribution of the Repetition Time to the reorganization of the network. A) LEFT Null distribution of the correlation coefficients between the PCC and DAS of the DMN. **RIGHT LEFT** Null distribution of the correlation coefficients between the DAS of the DMN and the lesion. **B) LEFT**, Correlations between node similarity measured by the Pearson correlation coefficient and the Dynamics Alteration Score (DAS) of the Default Mode Network (DMN; only for subjects with TR=2.4s). The inset shows how alterations in the dynamics shape the organization of the network regardless of the direction of the displacement of the frequency spectrum. **RIGHT**, Alterations in dynamics correlate with alterations in network complexity. Change in the complexity of the patients' DMN with respect to the healthy pool (Directionality $\Delta\theta$ and Magnitude $|\Delta\theta|$) as a function of the DAS (only for subjects with TR=2.4s). **C) LEFT**, Scatter plot showing the null correlation of DAS with distance and overlap between the lesion and the DMN centroids (only for subjects with TR=2.4s). **RIGHT**, A strong linear trend was found between alterations inside the patients' tumor and alterations in the DMN as measured by DAS (only for subjects with TR=2.4s). The orange line corresponds to the significant linear fit (two-sided exact test), and the shaded areas mark the 95% confidence interval.

Reviewer #2 (Remarks to the Author):

The authors have addressed all my comments and improved the manuscript significantly.

We appreciate the positive comments on the revised version following the suggestions.

Reviewer #3 (Remarks to the Author):

The authors are responsive and have addressed all my concerns.

We thank the reviewer for the previous suggestions.

REVIEWERS' COMMENTS:

Reviewer #1 (Remarks to the Author):

The authors have addressed all my concerns

Reviewer #1:

The authors have addressed all my concerns.

We thank the reviewer for the valuable suggestions. They have increased the replicability and transparency of our work.